# Intra-host changes in Kaposi sarcoma-associated herpesvirus genomes in Ugandan adults with Kaposi sarcoma

Jan Clement Santiago[1☉], Jason D. Goldman[2,3☉], Hong Zhao[1], Alec P. Pankow[1], Fred Okuku[4], Michael W. Schmitt[2], Lennie H. Chen[1], C. Alexander Hill[1], Corey Casper[2,3¤], Warren T. Phipps[2,3], James I. Mullins[1,2,5,6]*

1 University of Washington, Department of Microbiology, Seattle, Washington, United States of America, 2 University of Washington, Department of Medicine Seattle, Washington, United States of America, 3 Fred Hutchinson Cancer Research Center, Seattle, Washington, United States of America, 4 Uganda Cancer Institute, Kampala, Uganda, 5 University of Washington, Department of Global Health, Seattle, Washington, United States of America, 6 University of Washington, Department of Laboratory Medicine, Seattle, Washington, United States of America

☉ These authors contributed equally to this work.
¤ Current address: Infectious Disease Research Institute, Seattle, Washington, United States of America
* jmullins@uw.edu

**Data Availability Statement:** All whole KSHV genome sequence files are available from the NCBI, with GenBank Accession numbers: MT510648, MT510670, MT936340. All other relevant data are

## Abstract

Intra-host tumor virus variants may influence the pathogenesis and treatment responses of some virally-associated cancers. However, the intra-host variability of Kaposi sarcoma-associated herpesvirus (KSHV), the etiologic agent of Kaposi sarcoma (KS), has to date been explored with sequencing technologies that possibly introduce more errors than that which occurs in the viral population, and these studies have only studied variable regions. Here, full-length KSHV genomes in tumors and/or oral swabs from 9 Ugandan adults with HIV-associated KS were characterized. Furthermore, we used deep, short-read sequencing using duplex unique molecular identifiers (dUMI)–random double-stranded oligonucleotides that barcode individual DNA molecules before library amplification. This allowed suppression of PCR and sequencing errors to ~$10^{-9}$/base as well as afforded accurate determination of KSHV genome numbers sequenced in each sample. KSHV genomes were assembled *de novo*, and rearrangements observed were confirmed by PCR and Sanger sequencing. 131-kb KSHV genome sequences, excluding major repeat regions, were successfully obtained from 23 clinical specimens, averaging $2.3 \times 10^4$ reads/base. Strikingly, KSHV genomes were virtually identical within individuals at the point mutational level. The intra-host heterogeneity that was observed was confined to tumor-associated KSHV mutations and genome rearrangements, all impacting protein-coding sequences. Although it is unclear whether these changes were important to tumorigenesis or occurred as a result of genomic instability in tumors, similar changes were observed across individuals. These included inactivation of the K8.1 gene in tumors of 3 individuals and retention of a region around the first major internal repeat (IR1) in all instances of genomic deletions and rearrangements. Notably, the same breakpoint junctions were found in distinct tumors within single individuals, suggesting metastatic spread of rearranged KSHV genomes. These findings define KSHV intra-host heterogeneity *in vivo with greater precision than has been possible*

within the manuscript and its Supporting Information files.

**Funding:** This work was supported by National Institutes of Health (https://www.nih.gov) grants to U54 CA190146 (WTP), K23 CA 150931 (WTP) and the University of Washington Centers for AIDS Research Retroviruses and Molecular Data Sciences Core (P30 AI027757; JIM). The funders had no role in study design, data collection and analysis, decision to publish, or preparation of the manuscript.

**Competing interests:** I have read the journal's policy and the authors of this manuscript have the following competing interests: MWS is a founder and equity holder at TwinStrand Biosciences, Inc., which is commercializing Duplex Sequencing technology.

*in the past* and suggest the possibility that aberrant KSHV genomes may contribute to aspects of KS tumorigenesis. Furthermore, study of KSHV with use of dUMI provides a proof of concept for utilizing this technique for detailed study of other virus populations in vivo.

## Author summary

Kaposi sarcoma (KS) is a leading cancer in sub-Saharan Africa and in persons with HIV co-infection. Kaposi sarcoma-associated herpesvirus (KSHV, also referred to as human herpesvirus-8, or HHV-8) is the etiologic agent of KS, but the factors that contribute to the development of KS, which occurs in only a small subset of infected individuals, remain largely unknown. While strain differences or mutations in other tumor viruses are known to affect the risk and progression of their associated cancers, whether genetic variation in KSHV is important to the natural history of KS is unclear. Most studies of KSHV diversity have only characterized ~4% of its 165-kb genome, and the observed variation in some studies is likely to have been impacted by PCR or cloning artifacts. To precisely define genomic diversity of KSHV in vivo, we evaluated full-length viral genomes (except the internal repeat regions) using a technique that greatly lowers sequencing error rates and thus measures genomic diversity much more accurately than previous studies. In addition, we extended our analyses to look for potential tumor-specific changes in the KSHV genomes by examining viruses in both tumor and non-tumor tissues. To these ends, we performed highly sensitive, single-molecule sequencing of whole KSHV genomes in paired KS tumors and oral swabs from 9 individuals with KS. We found that KSHV genomes were virtually identical within the 9 individuals, with no evidence of quasispecies formation or multi-strain infection. However, KSHV genome aberrations and gene-inactivating mutations were found to be common in KS tumors, often impacting the same genes and genomic regions across individuals. Whether theses mutations influence KS tumorigenesis or result from genomic instability commonly found in tumors warrants further study. Lastly, aberrant KSHV genomes were found to be shared by distinct tumors within individuals, suggesting the capacity of KS tumor cells to metastasize and seed new lesions.

## Introduction

Kaposi Sarcoma (KS) is one of the most common cancers of HIV-infected individuals [1,2], with the burden of disease disproportionately borne by people in sub-Saharan Africa [3]. A gamma herpesvirus, Kaposi sarcoma-associated herpesvirus (KSHV), is the etiologic agent of KS and is consistently detected in tumor tissues [4,5]. KSHV is shed in saliva, which is thought to be a primary mode of transmission [6–8]. Only a small fraction of KSHV infections progress to KS, and the factors contributing to this progression are poorly understood. The development of KS is often associated with HIV infection and immunosuppression [9], but others factors, including KSHV genome variation, may contribute to differential outcomes of KSHV infection.

Studies of other human oncogenic viruses have revealed that viral genetic variation or *de novo* mutations may be important to their pathogenicity, as is the case for cancers associated with human papilloma viruses and Merkel cell polyomavirus [10]. Epstein Bar virus (EBV),

another gamma-herpesvirus like KSHV, is associated with a variety of neoplasms. In EBV infections, intra-host selection and compartmentalization of viruses may have a role in pathogenesis of EBV-associated cancers [11–13]. Additionally, EBV strains isolated from nasopharyngeal carcinomas (NPC) have unique genomic [14,15] and phenotypic [16–18] variations compared to other strains and isolates from geographically clustered individuals without cancer [19]. NPC-associated strains were also found to have increased tropism for epithelial- versus B-cells [16,20]. These findings suggest that viral genetic heterogeneity can affect EBV virulence. Whether KSHV genetic variation can similarly influence KS pathogenesis or manifestation is unknown.

KSHV encodes oncogenes that dysregulate cell cycle, cell-to-cell adhesion, inflammation and angiogenesis [21–23], and it is therefore plausible that heterogeneity in these genes might result in differences in KSHV infection outcomes and clinical manifestations. For example, polymorphisms in the microRNA (miRNA) region of KSHV have been correlated with the development of multicentric Castleman disease and KSHV-associated inflammatory cytokine syndrome with and without KS [24,25]. K1, the most variable KSHV gene, is conventionally used for KSHV subtyping. There are reports of certain K1 subtypes being associated with more aggressive KS [26–28], although correlations of KSHV genetic subtypes with worse KS clinical presentation or outcomes has not been consistently observed [29–32].

Studying whole KSHV genomes provides a far more comprehensive picture of diversity than the variable regions alone. Most publicly available KSHV genome sequences have been reported in only the last 5 years [33–37], and they demonstrate that polymorphisms in the ~130-kb KSHV non-repeat genomic region outside of the K1 gene contribute much more to KSHV diversity than the 0.9 kb hypervariable K1 gene by itself [33,34]. KSHV genomes have signatures of ancient recombination [36,38], resulting in mosaic genomes that could complicate disease risk association solely with K1 subtypes. It is possible that infecting KSHV strains may have similar pathogenicity, as principal component analysis of 70 KSHV whole genomes from saliva of different individuals with and without KS did not reveal strain specificity for KS [36]. However, further studies are needed to evaluate the possible impact of KSHV strain variation on KS development and clinical outcomes.

Whether KSHV infection commonly displays intra-host diversity is also unclear. Some studies examining virus in different anatomic sites or multiple isolates from an individual have reported detection of KSHV quasispecies, multi-strain infections [39–44] and intra-host KSHV viral evolution [37,42,45], while some studies of individuals with AIDS-associated KS have only reported a single persisting strain [46–49]. Apparent recombination events in the evolutionary history of KSHV [36,38,46,50,51] do suggest that co-infection of divergent KSHV strains occur, at least sporadically. However, limitations of commonly employed PCR technologies can undermine reliable interpretation of observed intra-host KSHV variation. For example, assessment of intra-host diversity can be easily biased by artifacts introduced during sample preparation and when sequencing PCR products [47]. Short read, next generation sequencing can also have high error rates due to PCR misincorporation, end-repair artifacts, insufficient sequencing depth, and DNA damage from long, repeated high-temperature incubations during PCR and enrichment reactions [52–55]. Application of newer and more sensitive sequencing technologies is needed to further reliably assess whether intra-host viral diversity is a common feature of KS.

To much more accurately assess minor intra-host KSHV sequence variation as well as tumor specific changes than has been done previously, we determined viral genome sequences in distinct anatomic compartments (oral and tumor sites), using a highly sensitive short-read sequencing method termed "duplex sequencing" [56]. This method incorporates duplex unique molecular identifiers (dUMI), which are double-stranded strings of random base pairs

used to barcode individual DNA molecules before PCR amplification and enrichment [56]. By utilizing dUMI-consensus reads of each DNA molecule in a sample library, PCR-associated errors are reduced to ~$10^{-9}$, revealing the original sequence variation within a sample [56]. In the present study, we report the results of successfully enriching and duplex sequencing whole KSHV genomes from tumors and oral swabs from 9 Ugandan adults with HIV-associated KS.

## Methods

### Ethics statement

All participants provided written informed consent. This protocol was approved by the Fred Hutchinson Cancer Research Center Institutional Review Board, the Makerere University School of Medicine Research and Ethics Committee (SOMREC), and the Uganda National Council on Science and Technology (UNCST).

### Study cohort and specimen collection

Specimens were obtained from participants enrolled in the "HIPPOS" Study, an ongoing prospective cohort study, begun in 2012, of KS patients initiating treatment at the Uganda Cancer Institute (UCI) in Kampala, Uganda. Participants were eligible for the HIPPOS study if they were >18 years of age with biopsy-proven KS, and naïve to antiretroviral therapy (ART) and chemotherapy at enrollment. Participants attended 12 study visits over a one-year period and received treatment for KS consisting of ART and chemotherapy (a combination of bleomycin and vincristine or paclitaxel) per standard protocols by UCI physicians. At each visit, participants received a detailed physical exam to assess clinical response using the ACTG KS response criteria [57].

Participants provided plasma samples at each visit for KSHV, CD4 and HIV viral load testing, and in addition, provided up to 12 biopsies of KS lesions before, during, and after KS treatment. KS tumor biopsies were obtained using 4mm punch biopsy tools after cleaning the skin with alcohol, and either snap-frozen at the clinic site and stored in liquid nitrogen (LN2) or placed in RNAlater (Sigma-Aldrich, Cat. # R0901) and stored at -80˚C. Study clinicians collected swabs of the oral mucosa at each study visit and participants self-collected oral swabs at home for 1 week after the visit after education on the sample collection technique, as previously validated by our group in Uganda [58]. Briefly, a Dacron swab is inserted into the mouth and vigorously rubbed along the buccal mucosa, gums, and hard palate. The swab is then placed in 1 mL of filter-sterilized digestion buffer [59] and stored at ambient temperature [60] before being placed at -20˚C for storage.

### DNA preparation

DNA was extracted from 300μL homogenized tumor lysates using the AllPrep DNA/RNA Mini Kit (QIAGEN, Cat. # 80204) and eluted into 100μL EB Buffer. For oral swab specimens, DNA was extracted from the swab tip eluate using the QIAamp Mini Kit (Qiagen, Cat. # 51304) following the manufacturer's protocol. Purification of DNA from saliva stabilized in RNAprotect Saliva Reagent (Qiagen) was performed following the manufacturer's protocol with the following modifications: There was no initial pelleting or PBS wash, 20 μL proteinase K was used per 200 μL specimen, and DNA was eluted in 50 μL water. DNA was quantified using a NanoDrop 1000 Spectrophotometer (ThermoScientific).

**PCR.** All PCR preparations were done in a PCR-clean room, except for the addition of control templates. PCR was conducted using the PrimeSTAR GXL kit (Takara, Cat. # R050B) with ThermaStop (Thermagenix) added. Cycling conditions were: 98˚C for 2 mins; 35 cycles

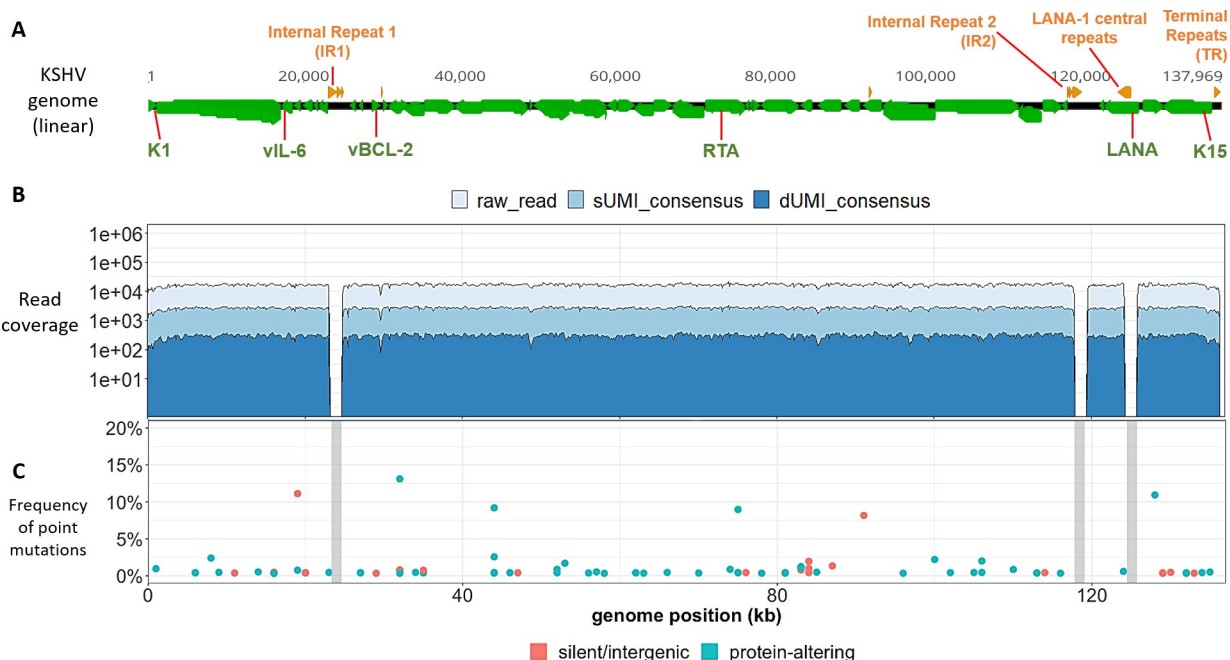

**Fig 1. KSHV genomes in BCBL-1 cells have low point mutational diversity.** (A) Schematic representation of a linear KSHV genome, with genes colored in green and the major repeat regions in orange. The locations of the K1, vIL-6, vBCL-2, RTA, LANA and K15 genes used for genome quantitation are indicated in green type. (B) Raw (light blue), sUMI-consensus (blue) and dUMI-consensus (dark blue) read coverage along the *de novo* assembled, BCBL-1 KSHV genome. (C) Bubble plot of minor sequence variants. Each bubble represents a position within the genome at which a variant base or indel was detected, colored by whether they were predicted to be silent or protein-altering mutations. Mutations likely to be silent included synonymous and intergenic point mutations, while protein-altering mutations included non-synonymous, nonsense and frameshift mutations. Bubble height represents variant base frequency among dUMI-consensus reads at that position. Vertical grey columns represent masked repeat regions in which no reliable alignments were possible.

of 98°C for 10 secs, 60–65°C (depending on primer) for 15 secs, 68°C for 1min/kb; 68°C for 3 mins and then hold at 4°C. Primer sequences are listed in **S1 Table**.

## Copy number quantification

KSHV genome copy numbers were quantified by droplet digital PCR (ddPCR) using the QX200 Droplet Generator and Reader (Bio-Rad), with ddPCR SuperMix for Probes (No dUTP) (Bio-Rad, Cat. # 186–3024). Primers and probes (**S1 Table**) were designed to detect 4 KSHV-unique genes K2/vIL-6, ORF16/vBCL-2, ORF50/RTA and ORF73/LANA (**Fig 1A**), with KSHV genome copy number reported as the average of the 4 measures. 420 ng BCBL-1 cell line DNA diluted 1:10,000 (~475 genome copies) was used as positive control, 1 ng human genomic DNA (Bioline, Cat. # BIO-35025) as negative control, and water as no template control. Cycling conditions were: 95°C for 10 mins; 40 cycles of: 94°C for 30 secs, 56°C for 30 secs, 60°C for 1 min; one cycle at 98°C 10 for mins, and then hold at 12°C. The KSHV on-target percent was calculated using the copy number quantification by ddPCR normalized to the total nucleic acid concentration.

## UMI-addition and library preparation (Fig 2)

To obtain ~500-bp DNA fragments, 10–20 ng/μL of DNA extract in 100 μL chilled TLE buffer (10mM Tris, pH8.0, 0.1mM EDTA) was sheared using a Bioruptor (Diagenode) on high power for up to 15 min. Fragment sizes were assessed on 1.5% agarose gels. Sheared DNA was bead-purified using 1.2X volume of Agencourt AMPure XP Beads (Beckman Coulter Cat. #

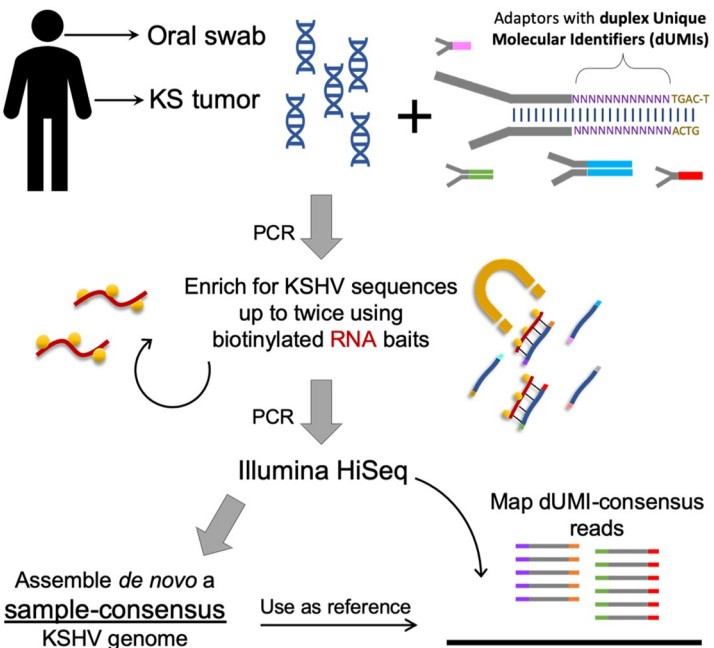

**Fig 2. Workflow for analyzing intra-host KSHV genome diversity from clinical samples.** Each study participant contributed multiple KS tumors and oral swabs, but only those samples with the highest viral loads were reported here. Sequencing libraries were prepared from DNA extracts from each sample with adaptors containing duplex Unique Molecular Identifiers (dUMIs, see **S1 Fig**). Adaptor-labelled DNA libraries were enriched using biotinylated RNA baits homologous to KSHV sequences. Captured DNA was PCR-amplified to levels sufficient for Illumina HiSeq sequencing. For most samples, libraries were subjected to a second round of enrichment and PCR amplification. Upon sequencing, whole KSHV genomes were first assembled *de novo* from each sample without the use of dUMIs. The sample-specific genomes generated (sample-consensus) were then used as reference to map the consensus of reads with identical dUMI-tags (i.e., dUMI-consensus reads).

A63880) and eluted in 50 μL water. Library preparation (end repair, A-tailing and adapter-ligation) was performed using the KAPA HyperPrep Library Preparation Kit (Cat. # KR0961/KK8503). Double-stranded DNA adapters contained a random 12-bp dUMI sequence and a defined 5-bp spacer sequence added to Illumina TruSeq adaptor sequences [61] (**S1 Fig**). Subsequently, DNA was bead-purified with 1X volume of beads and eluted in 50 μL water.

DNA libraries were subjected to pre-enrichment amplification with primers mws13 & mws20 (**S1 Table** and **S1 Fig**). PCR conditions were: 95°C 4 mins; 5–8 cycles of 98°C 20 sec, 60°C 45 sec, 72°C 45 sec; 72°C 3 mins, 4°C hold. If the bead-purified elution from the end repair and adapter step had more than 240 ng total, it was divided into 50 μL PCR reactions of ≤240 ng and pooled after amplification. PCR products were then bead-purified as above with 1.2X volume beads and elution in 100 μL water, quantified with Nanodrop, and their sizes assessed using a Bio-analyzer (Agilent DNA 7500) or Qiaxcel (QIAGEN AM420).

## Library enrichment & sequencing

Biotinylated RNA baits for enriching KSHV sequences in the library were those designed in [62] and were obtained from Agilent, Inc. (Santa Clara, CA). The design was a 120-bp, 12X tiling of the genome of KSHV isolate GK18 (Genbank ID: AF148805.2). The diversity of the bait library was further increased by including K1, ORF75, K15, ORF26 and TR sequences of strains JSC-1 (Genbank ID: GQ994935.1), DG1 (Genbank ID: JQ619843.1), BC-1 (Genbank ID: U75698.1), BCBL-1 (Genbank ID: HQ404500.1), Sau3A (Genbank ID: U93872.2), and of

all Western and African isolates in [29,33] (Genbank ID: AF130259, AF130266, AF130267, AF130281, AF130305, AF133039, AF133040, AF133043, AF133044, AF151687, AF171057, AF178780, AF178810, AF220292, AF220293, AY329032, KT271453-KT271468).

Target enrichment was performed using SureSelect Target Enrichment Kit v1.7 (Agilent) with all suggested volumes reduced by half. DNA hybridized to biotinylated-RNA baits was captured with streptavidin beads (Dynabeads MyOne Streptavidin T1, Invitrogen) and resuspended in 20μL water. The DNA-streptavidin bead mixture was used directly in post-enrichment PCR amplification with primers mws13 and mws21, the latter of which includes a sample index sequence (**S1 Table** and **S1 Fig**). The PCR cycle number ranged from 10–16, with products monitored every 2 to 3 cycles on a TapeStation (Agilent) to ensure correct fragment sizes (~500bp). When over-amplification resulted in library fragment sizes much larger than expected, a single "reconditioning"PCR cycle with fresh reagents was done [63]. PCR products were cleaned using 1.2X volume AMPure XP beads and the eluted DNA library was sequenced using Illumina HiSeq 2500 with 100-bp paired end reads. For some tumor samples with low KSHV copy numbers and all oral swab samples, a second library enrichment was performed.

### *De novo* assembly of sample-consensus genomes

Initially, a sample-consensus KSHV genome (**Fig 2**) was generated *de novo* from paired-end reads of each sample using custom scripts (**S2 Fig**, https://github.com/MullinsLab/ HHV8-assembly-SPAdes). At this stage, the first 17-bp from read ends were trimmed to remove dUMI sequences. Next, reads were subjected to windowed quality filtering using *sickle pe* [64] with a quality threshold of 30 and a window size 10% of read length. Filtered reads were aligned to a human genome (GRCh38 p12, GenBank GCA_000001405.27) using *bwa mem* [65]. Unmapped reads were used as input into the de novo assembler SPAdes v3.11.1 [66], with the setting *-k 21,35,55,71,81*. This often yielded 3 to 4 scaffolds that together encompassed the entire 131-kb unique sequence regions of KSHV, bounded by the major repeat regions: Internal Repeat 1 (IR1), Internal Repeat 2 (IR2), LANA central repeat and Terminal Repeats (TR) (**Fig 1A**). Next, all scaffolds over 500 bp were aligned using *bwa mem* to the genome of reference KSHV isolate GK18. From the aligned scaffolds a draft genome was generated in Geneious (Biomatters, Ltd) with manual correction as needed. To finish the assembly, GapFiller v1.1 [67] was used, setting *bwa* as the aligner and filtered paired-end reads as the input library. The genomes were annotated in Geneious based on the GK18 reference, also adding the annotation for long non-coding RNA T1.4 based on [68]. The major repeat regions were masked with Ns since they were poorly resolved by assembly of short reads that can map to multiple locations within the repeat regions.

### Variant identification from dUMI-consensus reads

Paired-end reads, including their dUMI sequence tags, were mapped by *bwa* to sample-consensus genomes (**Fig 2**) using a Makefile adapted from [61] (https://github.com/MullinsLab/ Duplex-Sequencing). Briefly, all reads mapping to the same genomic position were collapsed by single strand UMIs (sUMI) to make sUMI-consensus reads (**S2 Fig**). Complementary UMI tags from opposing strands were matched to create dUMI-consensus reads, thus removing nearly all PCR polymerase misincorporation and chimera artifacts. Nine bases from both read ends were then trimmed to minimize read end artifacts. Discrepancies between mapped dUMI-consensus reads and the sample-consensus genomes were manually inspected in Geneious and misalignments around homopolymer runs were corrected. Only the remaining discrepancies were considered to be sequence variants that existed prior to PCR amplification.

All genome and subgenome sequence alignments were generated using MAFFT [69] [algorithm FFT-NS-i x1000, scoring matrix 1PAM/k = 2], and all phylogenetic trees were inferred using RAxML [70] (-f d, GTR gamma, N = 100 starting trees), using a representative KSHV genome from each individual. The NeighborNet phylogenetic network was generated using SplitsTree5, excluding gap sites [71]. Consensus genome sequences were deposited in GenBank (Accession numbers MT510648—MT510670, MT936340, see **Table 1**) annotated with the genomic rearrangements, when present.

### Integration analysis

Systematic searches for KSHV integration into the human genome were done in two ways. First, each library was searched using local BLASTN against both human and KSHV sequences and then using the Perl script SummonChimera [72] to extract coordinates of potential integration sites. Second, a sample-consensus KSHV genome was appended as an extra chromosome to the human genome reference GRCh38 p12. The appended human genome reference was used to map sUMI consensus reads via Speedseq [73] to generate alignment files with only discordant or split reads. These were input into LUMPY for structural variant analysis [74]. Human chromosome sequences linked to KSHV sequences were taken to be putative integration sites.

## Results

### Assessment of the dUMI sequencing protocol with a KSHV infected cell line

As part of the optimization of the dUMI-sequencing protocol, KSHV genome sequences were first obtained from an early passage of BCBL-1, a KSHV-infected PEL cell line [75]. BCBL-1 cells were grown as previously described [76]. After DNA extraction, KSHV sequences corresponded to ~0.16% of the total DNA using a ddPCR assay for ORF73 and T0.7-K12, and normalized by comparison to the human gene POLG (DNA polymerase subunit gamma). Following a single round of bait capture, the fraction of sequence reads corresponding to KSHV from BCBL-1 DNA extracts (i.e., the "on-target" level), was 15.6%, corresponding to 173-fold enrichment.

Sequencing of the BCBL-1 KSHV genome produced a median coverage of 16,805 reads per base excluding the repeat regions. Collapsing raw reads by identical sUMI to generate sUMI-consensus reads resulted in a median of 2,677 sUMI reads per base. When collapsed further into consensus sequences derived from both strands, a median of 302 dUMI reads per base was obtained that were essentially free of PCR errors (**Table 1** and **Fig 1B**). Since each dUMI tags a unique DNA molecule before PCR, the number of unique dUMI tags indicates the number of unique viral templates sequenced [56,77]. Using this measure, 302 also approximated the number of KSHV genomes sampled from BCBL-1.

Eighty-one base positions (0.06%) in the BCBL1 consensus KSHV genome had detectable variants in dUMI-consensus reads, and the average frequency of minor variants was 1.35%. No variant exceeded 14% of the total dUMI-consensus reads at any position (**Fig 1C**). No doubling of read coverage was found within the 9-kb genomic region previously reported in the BCBL-1-derived KSHV recombinant clone BAC-36 [78].

The consensus, *de novo*-assembled KSHV genome in BCBL-1 had 3 differences from the published BAC-36 sequence: a C➔A change in the noncoding sequence before ORF K5 (BAC-36 position 24,630), 2 additional Gs in a homopolymer run at BAC-36 position 25,210), and a synonymous T➔C change in the K7 gene (BAC-36 position 28,409). No variant bases were found in dUMI-consensus reads at the equivalent positions of this BCBL-1 KSHV sequence,

**Table 1. Origin and processing results from specimens for KSHV genome analysis.**

| Pt ID | Age | Sex | Plasma HIV RNA (copies/mL) | CD4 T cells /µL | Sample ID | Sample Type | % on-target Pre-enrichment | % on-target Post-enrichment | Median read coverage | Standard Deviation | Median dUMI-consensus read coverage | Standard Deviation | Genome length (excluding masked repeats) | # positions with variants | Mean frequency of variant base | Genbank Accession |
|---|---|---|---|---|---|---|---|---|---|---|---|---|---|---|---|---|
| n/a | n/a | n/a | n/a | n/a | BCBL-1 | cell line | 0.0009% | 15.61% | 16,805 | 1,858 | 302 | 52.49 | 132,676 | 81 | 1.35% | MT936340 |
| U003 | 25 | M | 759,635 | 45 | U003-C | Tumor | 0.0037% | 7.15% | 16,265 | 1,715 | 270 | 30.47 | 131,292 | 21 | 1.80% | MT510648 |
| | | | | | U003-o1 | Oral swab | 0.0000% | 35.20% | 49,598 | 26,446 | 7 | 6.22 | 131,102 | 12 | - | MT510649 |
| | | | | | U003-o2 | Oral swab | 0.0003% | 31.60% | 66,527 | 13,995 | 310 | 68.48 | 131,129 | 218 | 0.51% | MT510650 |
| | | | | | U003-o3 | Oral swab | 0.0000% | 41.90% | 65,984 | 21,979 | 47 | 21.31 | 131,143 | 24 | - | MT510651 |
| U004 | 37 | M | 277,655 | 85 | U004-C | Tumor | 0.0016% | 17.20% | 19,582 | 3,078 | 34 | 7.41 | 131,277 | 12 | - | MT510663 |
| | | | | | U004-D | Tumor | 0.0017% | 75.10% | 60,340 | 10,253 | 1,291 | 280.27 | 131,237 | 73 | 0.73% | MT510665 |
| | | | | | U004-o1 | Oral swab | 0.0001% | 18.90% | 37,794 | 6,280 | 56 | 11.70 | 131,277 | 31 | - | MT510664 |
| U007 | 26 | M | 91,096 | 136 | U007-B | Tumor | 0.0006% | 86.90% | 75,001 | 19,771 | 1,172 | 268.44 | 131,352 | 156 | 0.17% | MT510654 |
| | | | | | U007-o1 | Oral swab | 0.0001% | 27.60% | 43,707 | 6,258 | 206 | 25.54 | 131,126 | 61 | 1.93% | MT510655 |
| U008 | 56 | M | 860,937 | 488 | U008-B | Tumor | 0.0049% | 16.97% | 19,309 | 4,168 | 637 | 148.42 | 131,142 | 31 | 0.93% | MT510656 |
| | | | | | U008-D | Tumor | 0.0040% | 29.98% | 26,209 | 4,674 | 195 | 42.95 | 131,102 | 24 | 0.70% | MT510657 |
| | | | | | U008-o1 | Oral swab | 0.0000% | 23.50% | 48,339 | 6,869 | 114 | 16.62 | 131,116 | 76 | 2.11% | MT510658 |
| U020 | 27 | M | 118,191 | 370 | U020-B | Tumor | 0.0070% | 1.47% | 2,913 | 14,971 | 24 | 196.60 | 131,102 | 10 | 2.40% | MT510666 |
| | | | | | U020-C | Tumor | 0.0003% | 34.38% | 420 | 3,674 | 3 | 2.51 | 131,471 | 30 | - | MT510667 |
| | | | | | U020-o1 | Oral swab | 0.0000% | 77.30% | 73,287 | 22,429 | 80 | 21.66 | 131,115 | 22 | - | MT510668 |
| U023 | 33 | F | 338,285 | 191 | U023-o1 | Oral | 0.0001% | 2.70% | 19,889 | 6,683 | 2 | 1.30 | 131,122 | 2 | - | MT510669 |
| U030 | 40 | M | 100,184 | 70 | U030-C | Tumor | 0.0135% | 15.68% | 38,935 | 6,664 | 490 | 62.59 | 131,282 | 17 | 1.08% | MT510670 |
| U032 | 23 | F | 587,149 | 274 | U032-B | Tumor | 0.0003% | 43.50% | 69,815 | 9,269 | 890 | 67.12 | 131,266 | 45 | 0.41% | MT510652 |
| | | | | | U032-o1 | Oral | 0.0000% | 7.70% | 7,311 | 3,218 | 1 | 0.80 | 130,842 | - | - | MT510653 |
| U034 | 47 | F | 130,375 | 237 | U034-B | Tumor | 0.0014% | 84.40% | 74,927 | 13,969 | 1,747 | 189.51 | 131,248 | 107 | 0.11% | MT510659 |
| | | | | | U034-C | Tumor | 0.0013% | 30.70% | 17,968 | 2,505 | 133 | 24.36 | 131,088 | 11 | 1.46% | MT510660 |
| | | | | | U034-o1 | Oral swab | 0.0000% | 7.30% | 7,851 | 2,818 | 2 | 1.22 | 130,754 | 2 | - | MT510661 |
| | | | | | U034-o2 | Oral swab | 0.0000% | 6.00% | 4,085 | 1,735 | 1 | 0.62 | 130,884 | - | - | MT510662 |

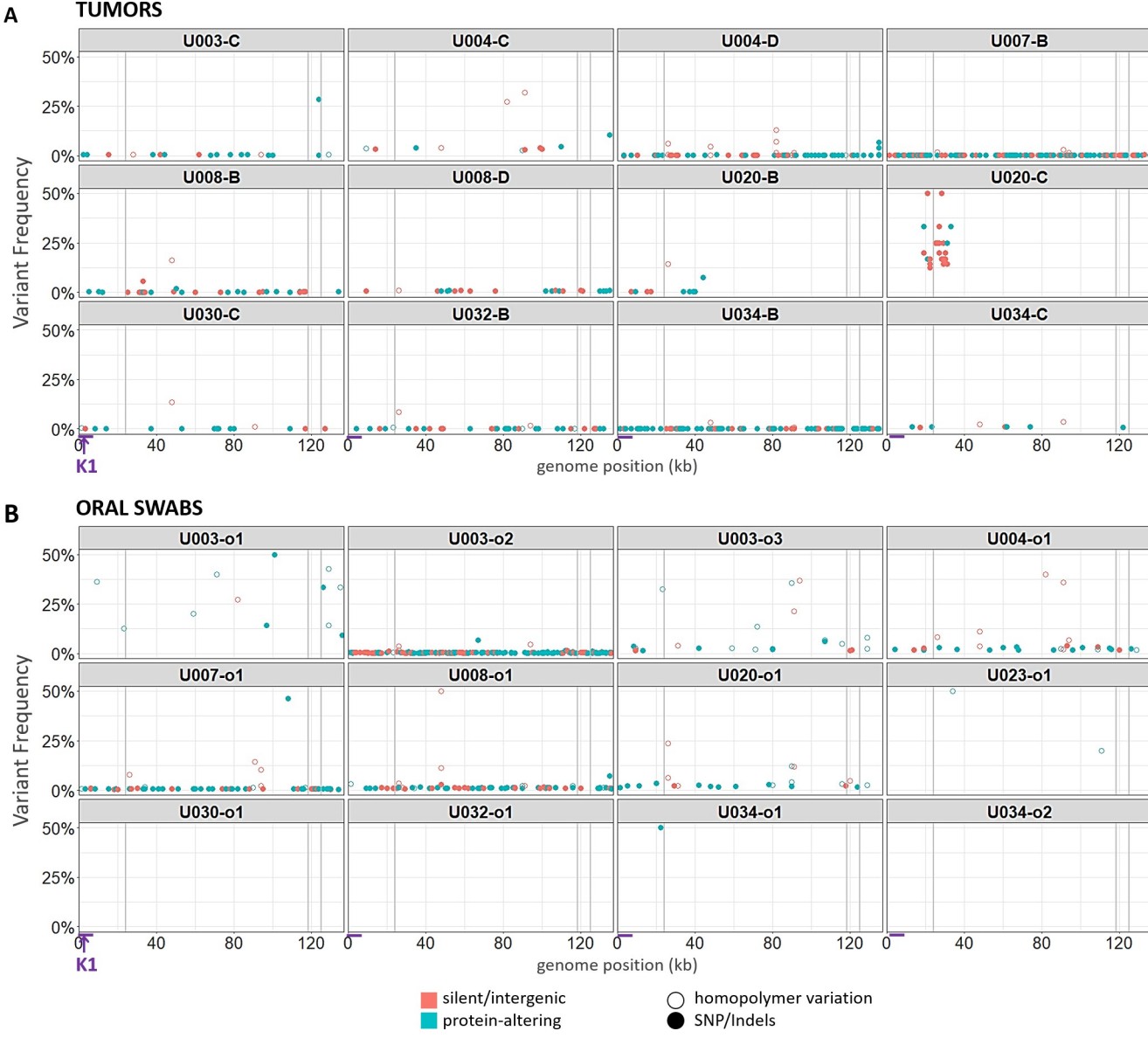

**Fig 3. Point mutational diversity in KSHV genomes from tumors and oral swabs.** Bubble plots of minor sequence variants remaining after removal of PCR errors, in KSHV genomes from tumors (A) and oral swabs (B). Each bubble represents a variant base or indel, colored by whether they were predicted to be silent or protein-altering mutations. Silent mutations include synonymous and intergenic point mutations, while protein-altering mutations included non-synonymous, nonsense and frameshift mutations. Hollow circles represent mutations occurring in homopolymer runs. Bubble heights represent the frequency of the variant base among dUMI-consensus reads at that position. Vertical gray columns represent the masked repeat regions. The region containing the K1 gene is indicated with arrows at the bottom of the figure.

indicating that the 3 BAC-36 sequence variants were not present in this passage of the BCBL-1 line at detectable levels (i.e., <1 copy per 302 genomes).

## KSHV sequence derivation from tumor tissues and oral swabs

KSHV genome sequences were successfully obtained from samples provided by 9 participants with HIV-associated KS, including 12 KS tumors and 11 oral swabs. (**Table 1**). The representation of KSHV DNA in a sample was determined by ddPCR analysis of KSHV genes vIL-6,

vBCL-2, RTA and LANA (**Fig 1A**) and provided as the percentage "on-target" KSHV DNA. These levels ranged from 0.03% to 1.35% (median 0.17%) in tumors, while most oral swab samples were below 0.01% on-target (**Table 1**). Following one enrichment with RNA baits, KSHV DNA corresponded to a median of 1.3% on-target, and after a second enrichment a median of 24.2% on-target, for a median final enrichment of 123,955-fold.

Median read coverage across KSHV genomes, excluding the major repeat regions, was 22,896 for tumors and 37,794 for oral swab samples. After collapsing mapped reads by dUMI, the median dUMI-consensus read coverage was 380 for tumors and 27 for oral swabs (**Table 1 and S3A and S3B Fig**). The lower dUMI-consensus read coverage of oral swab KSHV sequences, despite having higher raw read coverage than in tumors, was due to oral swab sample libraries having lower amounts of KSHV DNA and higher proportions of PCR duplicates. This resulted from low viral genome input requiring more rounds of enrichment and PCR cycles, and more significant DNA degradation during storage. Indeed, only 11 of 43 oral specimens attempted yielded acceptable library quality for whole KSHV genome sequencing, compared to 12 of 13 tumor samples. Since the median dUMI-consensus read coverage corresponds to the number of viral genomes sampled, tumor U032-B had the highest number of genomes analyzed at 1,653. We set the lowest number of genomes accepted for confident assignment of variant frequencies to be 100 (**S4A Fig**); below this number dUMI-consensus read coverage was judged to be too sparse. U020-B was an exception due to most genomes having a large deletion, to be discussed below. For other samples with sequenced genomes below 80, dUMI-consensus reads generated were insufficient to cover the entire KSHV genome, even if whole KSHV genomes could be assembled from raw reads. Overall, read coverage was relatively uniform along the KSHV genome for most tumors (**S3A Fig**) and all adequately sampled oral swabs (**S3B Fig**).

Very few point mutations were found in dUMI-consensus reads from either tumors (**Fig 3A**) or oral swabs (**Fig 3B**). Excluding the major repeat regions, the number of genome positions with a detectable intrasample variant base ranged from 2–218 (<0.01–0.17%) (**Table 1**). These frequencies were lower or comparable to those in the BCBL-1 cell line, although clinical samples had detectable variation in long homopolymer runs not observed in the BCBL-1 viruses. The sample-consensus genome was generally the only KSHV sequence present in each sample, hence, there was no evidence for the existence of quasispecies [79].

Artifacts resulting from the end-repair step in DNA library preparation, which precedes the application of dUMI tags, cannot be corrected by duplex sequencing [56,61,80]. Hence, 9 bases were trimmed from ends of dUMI-consensus reads before analyses, substantially reducing the variation observed in the raw reads. The minor base variants remaining in all samples revealed a preponderance of C➔A and G➔T substitutions (**S4B Fig**) as well as differences in homopolymer run lengths (**Fig 3A and 3B**). However, most minor variants were supported by only one dUMI-consensus read. Overall, mean variant frequency and median dUMI-consensus read coverage were inversely correlated (**S4A Fig**). Since the remaining variation cannot be distinguished from artifacts, true minor variant frequencies could be even lower than reported here.

## KSHV genomes were virtually identical at the point mutational level between tumors and oral swabs from the same individual

Intra-individual single nucleotide differences between tumor and oral swabs ranged in number from 0–2 across entire ~131-kb genomes, not counting the major repeat regions. Notably, there were almost no intra-individual polymorphisms in the KSHV hypervariable gene K1 (**Fig 3A and 3B**). Hence, no evidence for minor KSHV variants or multi-strain infections was found in these individuals.

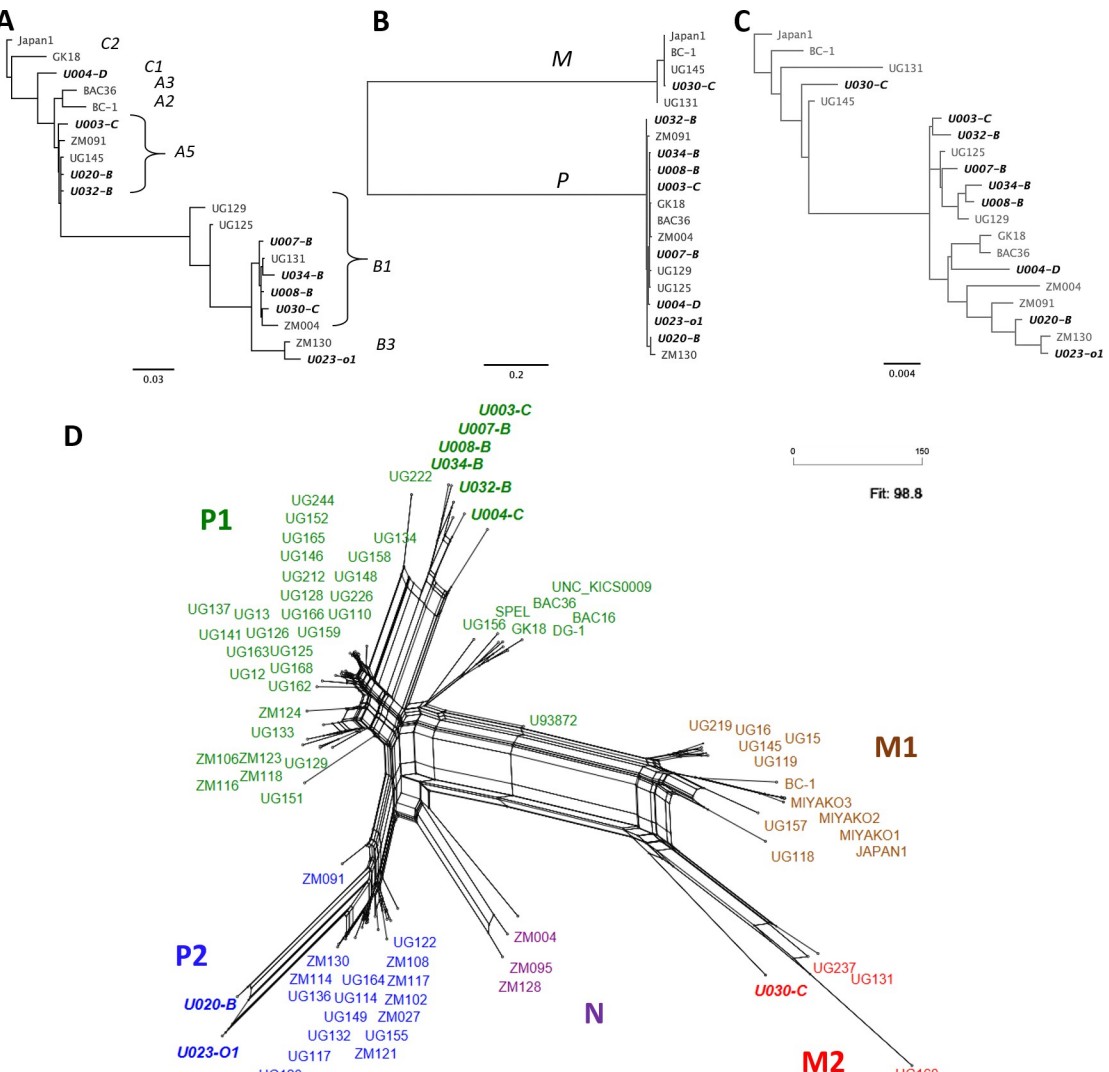

**Fig 4. KSHV phylogenetic relationships in variable regions K1 and K15 and whole genomes.** Phylogenetic trees of (A) K1 genes, (B) K15 genes and (C) whole genomes from this study and of select genomes from other publications. K1 and K15 subtypes are indicated in the K1 (A) and K15 (B) trees. (D) A neighbor-net phylogenetic network of all published KSHV genomes to date, color-coded by genome types proposed in [36]: P1 in green, P2 in blue, N in purple, M1 in brown and M2 in red. All *de novo*-assembled genomes from this study are in bolded italics.

KSHV genomes were distinct across the 9 participants, with sequence differences ranging from 3.06–4.85%. They included K1 subtypes A5, B1 and C3 (**Fig 4A**) and K15 alleles P and M (**Fig 4B**). While K1 and K15 are the most variable KSHV genes, polymorphisms along the rest of the genome have been reported to contribute more in aggregate to the total diversity of KSHV [33,34,36]. Consistent with this, maximum-likelihood phylogenetic trees using entire KSHV genomes (**Fig 4C**) were topologically distinct from those of K1 or K15. Moreover, due to signatures of recombination in the evolutionary history of KSHV [36,38], differing phylogenies along sections of the KSHV genome may be better represented by a phylogenetic network (**Fig 4D**), in which higher degrees of conflict result in a more web-like structure rather than a tree.

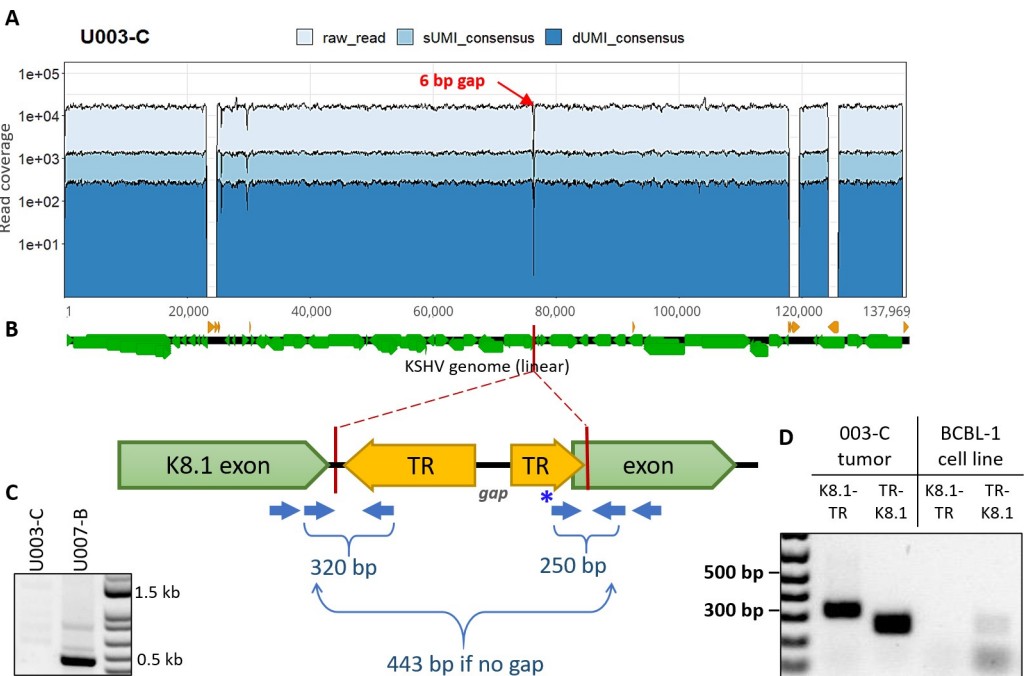

**Fig 5. KSHV genomes in the U003-C tumor harbor a deletion within the K8.1 gene.** (A) Read coverage of the U003-C KSHV genome, showing a 6-bp gap (red arrow) where no read pairs were mapped. (B) Cartoon of the *de novo*-assembly sequences generated at either side of the gap, both ended within the K8.1 gene intron and continued into terminal repeat (TR) sequences. Green and yellow arrows show the directions of the K8.1 gene and terminal repeat sequences, respectively. Blue arrows show the position of PCR primers used to confirm breakpoint junctions, with the expected PCR product sizes. (C) PCR products generated from U003-C tumor DNA using primers flanking the gap. The 443-bp PCR product expected if the K8.1 gene intron was intact was not detected from U003-C (left column), and was detected in tumor U007-B (right column) from another person. (D) Hemi-nested PCR of U003-C tumor DNA for the K8.1-TR (left) and TR-K8.1 (right) junctions produced products of the predicted sizes. These structures were confirmed by Sanger sequencing. No K8.1-TR or TR-K8.1 junction fragment was produced from BCBL-1 DNA. The light bands at the TR-K8.1 lane under BCBL-1 were determined from Sanger sequencing to be amplicons generated from the forward primer sequence (indicated with * in panel B) overlapping with K8.1; this primer was used since the rest of the connected TR sequence assembled was GC-rich and unsuitable for primer design.

## Aberrant KSHV genome structures in tumors

Among the 12 tumor-derived KSHV genomes examined, 7 had anomalous read coverage that shifted abruptly once or twice along the viral genome (**S3A Fig**). In contrast, oral swab KSHV genomes from the same individuals had uniform read coverage while being identical in sequence. This argues against preferential target capture by RNA baits or sequencing biases. Repeating the enrichment and sequencing, conducted on some samples, reproduced their distinctive read coverages. Split reads accumulated at the points of abrupt shifts in read coverage, even after collapsing reads by their dUMI consensus, indicating that these were not PCR template switching artifacts. Specific viral genomic anomalies observed are detailed below, along with any additional evidence showing that they represented real structural aberrations in viral genomes:

**Tumor U003-C.** Read coverage in U003-C was high (average of 15,635) and uniform across the KSHV genome except for a 6-bp gap within the K8.1 gene intron up to the first base of the second K8.1 exon (**Fig 5A**). No read indicated a deletion, nor was any read found with its mate pair located across the 6-bp gap. This region was PCR amplified from unsheared U003-C tumor DNA using conserved primers flanking the gap (**Fig 5B**), and no PCR product was detected. In contrast, an intact K8.1 intron sequence was amplified and sequenced from a tumor from another person (**Fig 5C**).

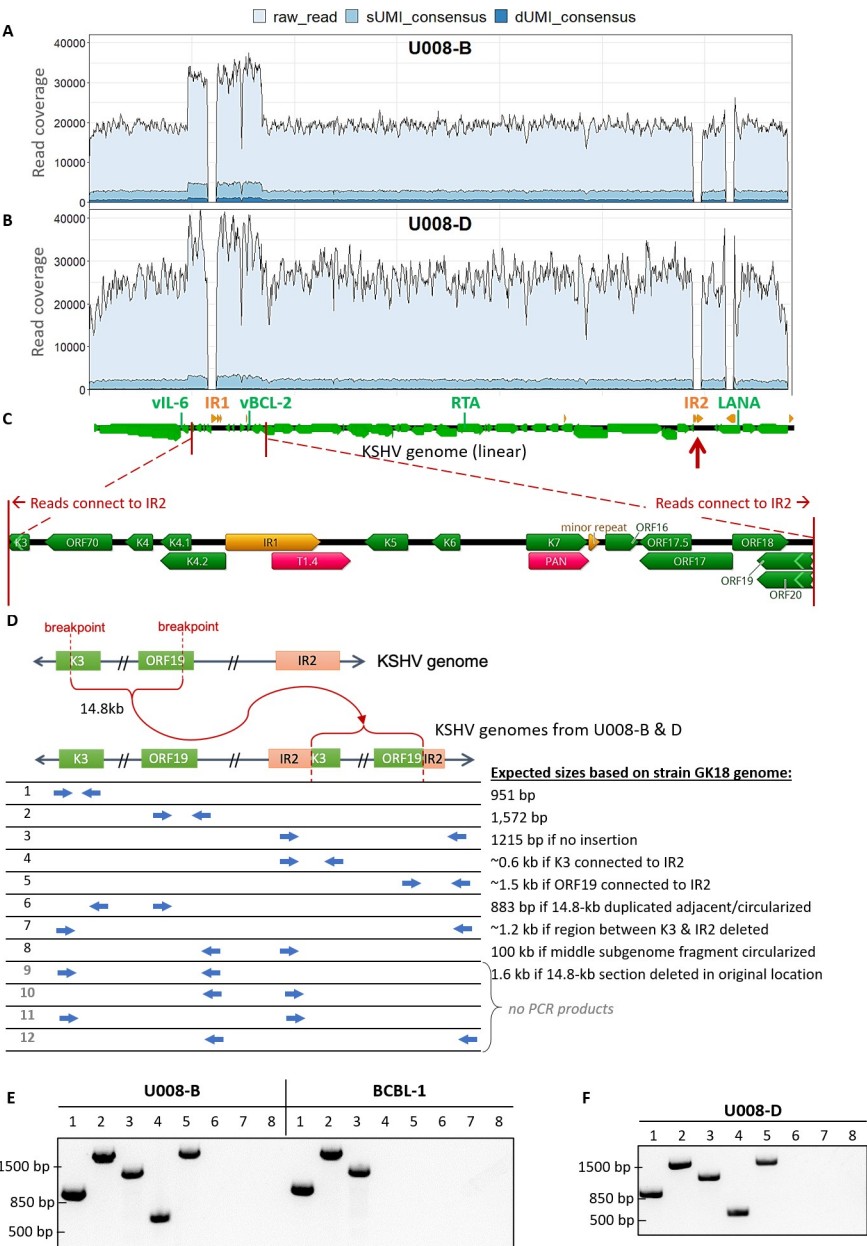

**Fig 6. KSHV genomes in two tumors from participant U008 had a 14.8-kb region flanking Internal Repeat 1 (IR1) duplicated and translocated to Internal Repeat 2 (IR2).** Total, sUMI and dUMI-consensus read coverage of tumor B (A) and D (B) genomes from individual U008. (C) Annotations of the region with 1.5-2X read coverage, with genes in green, repeat regions in orange, and long non-coding RNAs in red. Many reads on the edge of this region continue into IR2 (red arrows). Annotations are from the KSHV reference isolate GK18. (D) Cartoon showing the duplication of the 14.8 kb region into IR2 and the PCR primers used to examine the genomic rearrangement in unsheared tumor DNA extracts from tumors U008-B and U008-D. PCR products produced from primer pairs numbered in D from U008-B and BCBL-1 (E) and in U008-D (F). All visible bands were excised from the agarose gel and sequenced, confirming the indicated junction sequences. Primer pairs # 9–12 produced no PCR products discernible on an agarose gel.

*De novo* assembly revealed that the reverse complement of TR sequences continued from the gap position (**Fig 5B**). K8.1-TR junctions were confirmed by PCR with primers flanking the junctions (**Fig 5D**) and Sanger sequencing. Inversion of the 60-kb 3' end of the U003-C genome, starting inside K8.1, is a parsimonious explanation for the breakpoints.

**Table 2. Gene copy numbers in tumor or oral swab DNA.**

| Sample | vIL-6 | vBCL-2 | RTA | LANA |
|---|---|---|---|---|
| U003-C | N/A | N/A | N/A | 9,664 |
| U003-o1 | 127 | 100 | 120 | 135 |
| U003-o2 | 1,298 | 1,238 | 1,452 | 1,628 |
| U003-o3 | 191 | 184 | 169 | 199 |
| U004-C | 1,243 | 1,274 | 1,205 | 998 |
| U004-D | 4,433 | 4,466 | 4,543 | 4,290 |
| U004-o1 | 117 | 128 | 120 | 112 |
| U007-B | 1,842 | 1,864 | 2,040 | 1,925 |
| U007-o1 | 376 | 356 | 322 | 344 |
| U008-B | 19,140 | 33,629 | 19,910 | 19,195 |
| U008-D | 24,360 | 34,755 | 12,737 | 17,189 |
| U008-o1 | 129 | 136 | 119 | 138 |
| U020-B | 49,600 | 55,850 | 4,550 | 5,500 |
| U020-C | 3,658 | 5,033 | 145 | 139 |
| U020-o1 | 254 | 231 | 234 | 248 |
| U020-o2 | 100 | 183 | 123 | 225 |
| U023-o1 | 62 | 57 | 50 | 66 |
| U030-C | N/A | N/A | N/A | 59,730 |
| U030-o1 | 2 | 5 | 5 | 7 |
| U032-B | 476 | 520 | 494 | 466 |
| U032-o1 | 9 | 0 | 2 | 0 |
| U034-B | 1,920 | 1,887 | 1,870 | 1,793 |
| U034-C | 13,083 | 13,335 | 8,148 | 6,552 |
| U034-o1 | 9 | 6 | 9 | 7 |
| U034-o2 | 6 | 11 | 0 | 1 |

N/A, not quantified.

**Tumor U004-D.** The first 3kb, from K1 to the end of gene ORF4, had 1.5X read coverage compared to the rest of the KSHV genome (**S3A Fig**). However, no split reads or chimeric read pairs were found to explain this result from a genome rearrangement or deletion.

**Tumor U008-B and D.** U008-B had 1.7X greater read coverage over a 14.8-kb segment from inside K3 to inside ORF19 (GK18 reference positions 19,168 to 33,980, **Fig 6A**), including IR1 (masked). This was corroborated by ddPCR quantitation of vBCL-2, inside the 1.7X coverage region, with 1.7–1.9-fold higher gene copy number in the tumor compared to the vIL-6, RTA and LANA regions (**Table 2**).

Inferring from split reads, the 14.8-kb segment was translocated to inside IR2 (to GK18 position 119,496, **Fig 6D**). This was confirmed in the unsheared tumor DNA by PCR and Sanger sequencing using primers spanning the breakpoint (**Fig 6D and 6E**, lanes 4 and 5). Other primer pair combinations were tested to see if there were DNA species with the 14.8-kb segment inverted, deleted in place, duplicated in tandem or rearranged in other ways. None generate detectable PCR products except for primer pairs showing that the 14.8-kb segment also exists in the native configuration (**Fig 6E**). Thus, the 14.8-kb segment was copied into IR2 but had not been deleted from its original location.

In a parallel study of viral transcriptomes [81], abundant expression of a chimeric Kaposin transcript fused to the same 14.8-kb segment was found in tumor U008-B, consistent with the

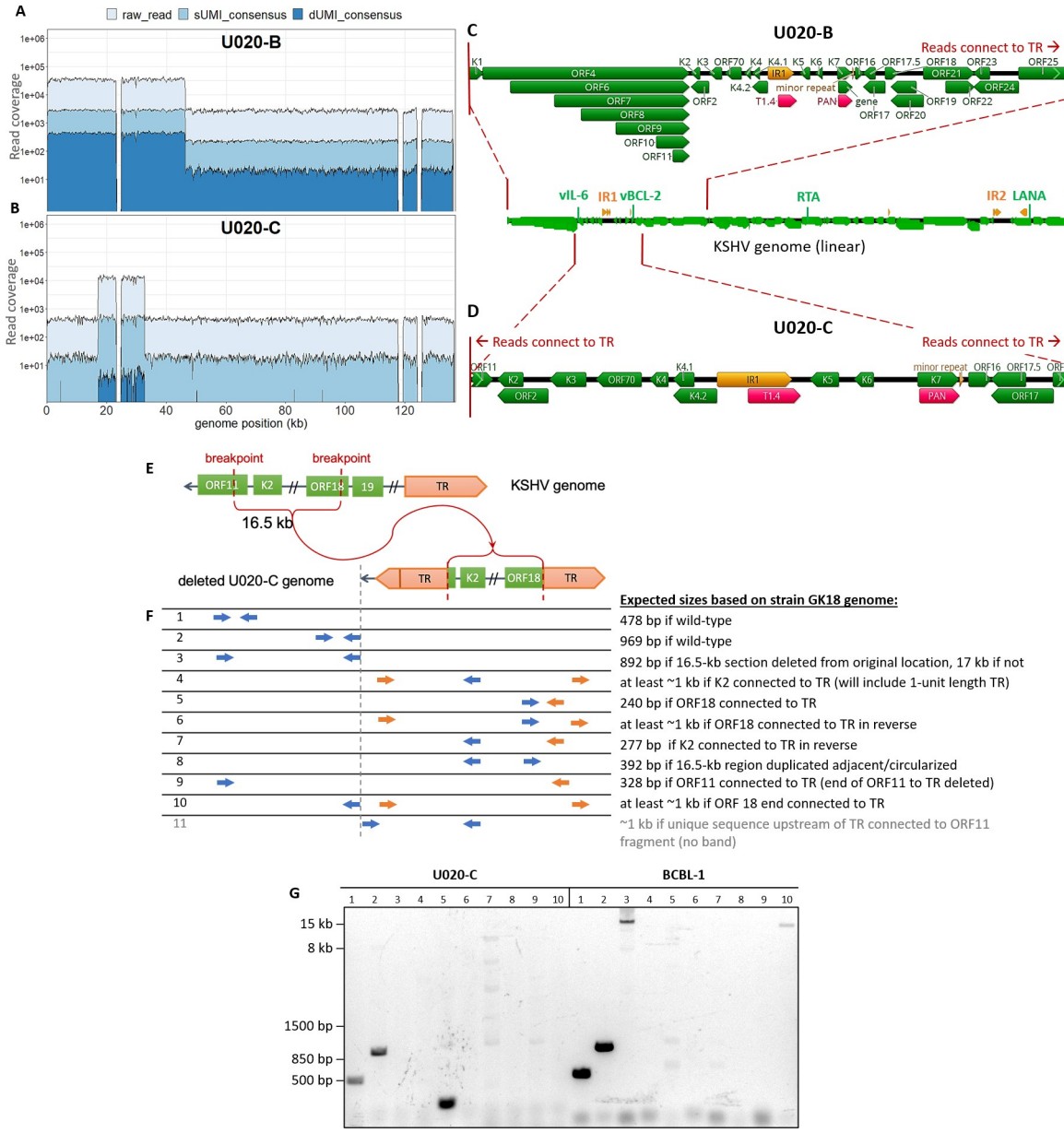

**Fig 7. KSHV genomes in U020-B and U020-C have large, distinct deletions.** Total, sUMI and dUMI-consensus read coverages of U020-B (A) and U020-C (B) KSHV genomes. GK18 reference annotations in the high-coverage regions of U020-B (C) and U020-C (D). (E) Cartoon showing the region encompassing the high coverage region of U020-C viral genomes, leaving a 16.5-kb region connected to TR. (F) PCR primers used to examine genome structure. Primers to unique genomic sequences are in blue, and primers to repeat sequences are in orange. (G) PCR products produced from primer pairs numbered in E, with DNA from U020-C and BCBL-1 as templates. Bands from lanes 1, 2 and 5 were excised from the agarose gel and sequenced. Faint bands in lanes 7 (five bands) and 9 (one) under U020-C were extracted from the gel but did not yield enough DNA for Sanger sequencing, except for the light 1-kb band in lane 7. Top BLAST hits to this sequence were human phosphatidylserine synthase 2 gene sequences, likely amplified due of spurious primer homology. The ~16 kb band in lane 3 under BCBL-1 was confirmed by Sanger sequencing to be ORF11 and ORF18 sequences from its two ends, and lane 10 under BCBL-1 corresponded to sequences common to many cloning vectors such as pCMV-VEE-GFP. Row 11 primers in (F), in which the forward primer binds to unique genomic sequences preceding the TR, yielded no discernible product.

viral genome structure we observed. Another tumor from the same participant, **U008-D** (**Fig 6B**), had 100% nucleotide identity and was confirmed to have the same duplication and break-point junctions (**Fig 6F**).

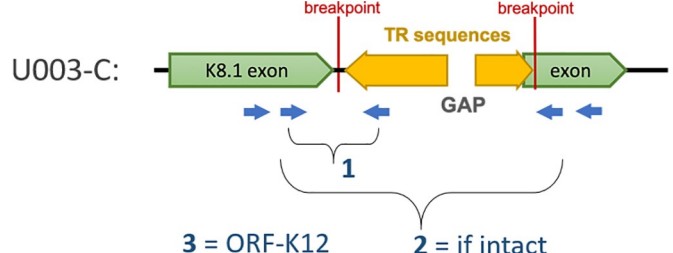

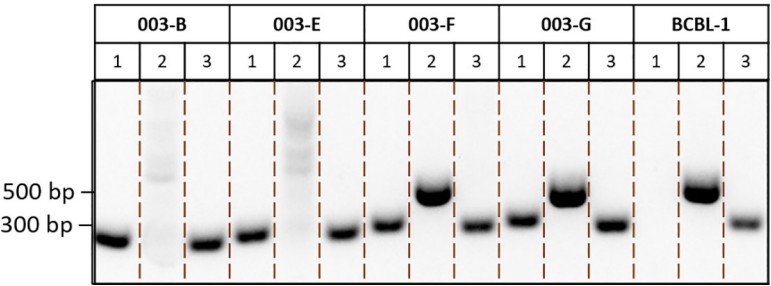

**Fig 8. KSHV genome structures in participant U003.** Junction sequences marking the genomic aberration in U003-C were detected in all 4 other tumors tested, while intact K8.1 sequences were detected in only 2. The cartoon shows the breakpoints in the K8.1 intron of U003-C extending into TR sequences, along with PCR primers used to confirm the genome structure. PCR products from other tumors of participant U003 and from the BCBL-1 cell line are shown at the bottom of the figure. All visible bands were excised from the agarose gel and their structures confirmed by Sanger sequencing.

**Tumor U020-B.** Read coverage abruptly dropped 12.8-fold over the last ~90 kb of the KSHV genome in this tumor (**Fig 7A**). This was consistent with ddPCR quantitation, with vIL-6 and vBCL-2 gene amplicons having 9.0–12.3-fold higher levels than RTA and LANA (**Table 2**). The coverage shifted before the end of ORF25 (GK18 position 46,615) and reads at this breakpoint continue into TR sequences ~90 kb downstream (**Fig 7C**). Thus, U020-B appeared to have KSHV genome variants with a ~90-kb deletion, or formally, a 12.8X amplification of a 46-kb subgenomic region. No U020-B tumor DNA remained to allow confirmation of this breakpoint.

**Tumor U020-C.** This tumor had a 30-fold difference in read coverage and breakpoints inside ORF11 and ORF18 (**Fig 7B**). ddPCR quantitation demonstrated gene copy numbers of vIL-6 and vBCL-2 amplicons to be 25–36-fold higher than for RTA and LANA (**Table 2**). The spike in read coverage occurred over a 16.2 kb region (GK18 positions 16,942 to 33,011). Again, chimeric reads were found at either ends of this region continuing into TR sequences (**Fig 7D**), indicating fusion with the TR (**Fig 7E**). Junction fragment-specific PCR and Sanger sequencing confirmed the 3' junction (**Fig 7G**, lane 5). No PCR product was produced from the other putative breakpoint junction, TR-ORF11 (**Fig 7F**, lane 4 primers in K2 and TR). However, the latter result is likely due to failure of PCR priming to the extremely GC-rich TR sequences, which was largely unsuitable for PCR priming [82]. Three forward primer designs paired with the reverse TR primer that had worked for the 3' junction (**Fig 7G**, lane 5), produced no PCR products even from BCBL-1 DNA. Finally, no product was produced from primer pairs outside and flanking the 16.5kb region (**Fig 7G**, lane 3), as well as with a primer binding upstream of TR (**Fig 7G**, lane 11), but these were most likely due to these primers being complementary to the deleted region (<3 reads per base). Determining the presence of

>16 kb DNA from very low template amounts requires very long PCR extension times and high template numbers and were beyond the limit of our PCR assays.

**Tumor U030-C.** Greater than 30,000 reads/position were uniformly observed throughout most of the KSHV genome. However, coverage dropped or was missing within the K15 gene (**S3A Fig**). The remaining K15 sequences corresponded to the K15 M-allele, which is less common than the P allele but was included in our RNA bait design (GenBank U75698). PCR amplification and Sanger sequencing of this region showed that the U030-C tumor did contain some copies of the entire M-allele K15 sequence. The U030-C sample-consensus genome was finished with this Sanger sequence result, since no dUMI-consensus reads mapped to the gaps in K15. In the parallel RNAseq study of tumors of this same participant, U030-B and C, transcripts of K15 were also lacking, unlike tumors from all other participants [81].

## The same aberrant KSHV genomes are found in multiple lesions from the same individual

In the case of U008-B and U008-D, two tumors biopsied from distinct lesions on the left leg (**S5 Fig** and **S2 Table**), full-length genome sequencing showed that they had the same 14.8 kb KSHV subgenomic sequence duplicated in IR2 (**Fig 6F**). PCR primers across those breakpoints were used to screen for the same structures in 6 other distinct lesions (**S2 Table**) from this individual, and none had this duplication. In contrast, four additional tumors tested from participant U003 had the same inversion breakpoints, detected by PCR, as tumor U003-C (**Fig 8**). Moreover, no intact K8.1 sequences were detected in 2 of these 4 tumors by nested PCR of the region spanning the K8.1 intron gap (**Fig 8**). These biopsies came from distinct lesions in the left leg (**S2 Table**). Lastly, in participant U020, the ORF18-TR junction sequences found spanning the U020-C genomic deletion was not detected in the 2 other tumors tested.

## Mutations in sample-consensus KSHV genomes from tumors impacted protein coding sequences

Among the 7 participants with KSHV sequences from at least one oral swab and one tumor, sample-consensus KSHV genomes were identical in the oral and tumor samples of 2 participants and had a few differences in 4 others. In the remaining participant, U004, the sample-consensus KSHV genome in one tumor was identical to that in oral but the second tumor had mutations. The mutations unique to tumors were typically nonsynonymous point mutations resulting in highly dissimilar residues or were mutations that might disrupt their expression (**Table 3**).

Several of the mutations or genome aberrations observed in tumors occurred in structural genes (**S3 Table**), and frequently involved the K8.1 gene, which encodes an envelope glycoprotein: The U003 inversion breakpoint cleaved the K8.1 gene; U004-D had an R56Q mutation in its ORF32 tegument protein coding sequence, as well as a 28-nt deletion in the promoter region of K8.1 (**S6A Fig**). The deletion was after the K8.1 core promoter sequence [83], but encompassed the K8.1 transcription start site [84]; the ORF25 major capsid protein in U020-B had a Q594K mutation, in addition to the U020-B genomic deletion that started further downstream inside ORF25; U020-C had a nonsense mutation at the beginning of the second K8.1 exon; and U032-B had a T848A mutation in ORF63, a tegument protein.

The only intra-host synonymous point mutation observed was in ORF K12 of U003-C (GK18 position 118,082). This C to T change occurred within the oncogenic microRNA K10 (miR-K10a-3p) sequence embedded in the K12 transcript. The three oral swab samples from this participant maintained the consensus C at this position (**S6B Fig**), whereas the 4 other tumors from this participant examined had T at this position, with tumor U003-G having a

**Table 3. Unique KSHV mutations observed in tumors compared to oral swabs from the same individual.**

| Sample ID | Tumor-specific differences |
|---|---|
| **U003**-C | K12 synonymous mutation within miR-K10 |
| | genomic inversion starting at K8.1 |
| **U004**-C | NONE |
| **U004**-D | ORF32 nonsynonymous mutation R56Q |
| | K15 nonsynonymous mutation A290P |
| | 28-nt deletion within the K8.1 promoter |
| | 3-kb segment duplication from before K1 to after ORF4 |
| **U007**-B | NONE |
| **U008**-B | duplication of 14.8 kb segment around IR1 into IR2 |
| | - breakpoints inside K3 & ORF19 |
| | - genes duplicated: ORF70, K4.1, K4.2, K5, K6, K7, ORF16, ORF17, ORF17.5, ORF18 |
| **U008**-D | same as U008-B |
| **U020**-B | ORF25 nonsynonymous mutation Q594K |
| | genomic deletion connecting end of ORF25 coding sequence to TR sequences, 47 kb remaining |
| **U020**-C | ORF11 nonsynonymous mutation T396P |
| | K3 nonsynonymous mutation F88L in transmembrane domain |
| | K8.1 nonsense mutation at start of 2nd exon |
| | genomic deletion leaving only 16 kb segment surrounding IR1 connected to TR sequences |
| | - breakpoints inside ORF11 and ORF18 |
| | - ~30X coverage for: K2, ORF2, K3, ORF70, K4, K4.1, K4.2, K5, K6, K7, ORF16, ORF7, ORF17.5 |
| **U032**-B | ORF63 nonsynonymous mutation T848A |
| **U034**-B | NONE |
| **U034**-C | NONE |

minor population of viruses with the consensus C (**S6C Fig**). The change was outside the seed sequence of miR-K10a-3p and may have resulted in a slightly more stable stem loop precursor (ΔG -32.40 ➡ -32.70, **S7 Fig**).

## Lack of evidence for integration of KSHV sequences into human chromosomes

No *de novo*-assembled scaffolds, split reads or improperly-paired read mappings suggested any instance of KSHV sequences fused to human DNA. Nevertheless, attempts were made to systematically search for human-KSHV chimeric sequences. The methods employed were the same as those used to screen for all integrated herpesviruses sequences in public databases [85] and EBV integration sites in in primary gastric and nasopharyngeal carcinomas [86]. The KSHV genome inversions, duplications and deletions described above were detected with high confidence values. In contrast, putative breakpoints joining human and KSHV sequences were supported by only tens of reads, about 2 orders of magnitude lower in number, and often involved LANA repeats into low-complexity human repeat sequences. These were judged to be artifacts.

## Co-infection with EBV detected predominantly in oral swabs

Some scaffolds during *de novo* assembly correspond to EBV sequences. Nearly all oral swab samples yielded multiple EBV-mapping scaffolds up to 73 kb, with no region of the EBV genome over-represented. In contrast, EBV-sequences were detected in only 5 of 12 tumors, and in all cases were sequences flanking the EBNA-1 repeat. The proportion of reads mapping

to EBV in oral swabs ranged from 0–33%, median 1.8%, whereas in tumors the range was from 0–0.5%, median 0.002% (**S4 Table**). No other eukaryote viruses were identified.

## Discussion

This study is the first to explore KSHV intra-host diversity at the whole genome level, across tumor and non-tumor sites, and provides an unprecedented level of precision to KSHV genome sequence analysis definition in clinical specimens. KSHV genome sequences were obtained essentially error-free from 11 oral swabs and 12 KS tumor samples provided by 9 Ugandan adults with KS. More than 100 genome copies per sample were sequenced from most tumors, while fewer copies were sequenced from oral swabs due to lower input and sample quality. By incorporating dUMI, PCR misincorporation errors and template switching artifacts were substantially eliminated, permitting detection of variants as infrequent as 0.01% and a theoretical error rate of $1/10^{-9}$, approximately the DNA replication error rate in eukaryotic cells [56]. Given that herpesviruses have among the lowest replication error rates among viruses [87], next-generation sequencing errors would have corresponded to a greater proportion of suspected mutations and confounded discernment of any rare variants in a potentially heterogeneous mixture, regardless of read depth [88].

We did not observe evidence of KSHV quasispecies within individuals [79], consistent with the low mutation rates of large dsDNA viruses [87]. Less than 0.01% of all base positions in the 131-kb KSHV genomes (excluding the major repeat regions), or <1 site per genome, were found to have a detectable variant, and these were typically supported by only one dUMI-consensus read. While the possibility of artifacts cannot be totally excluded, the exceedingly low intra-sample variation observed is within the published resolution of dUMI sequencing [56]. There are reports of intra-host KSHV variability in certain KSHV-endemic populations [39], in children [45], in iatrogenic settings [41–43] and in blood of AIDS-KS patients [44], but these findings were arrived at by Sanger sequencing of PCR amplicon clones of hypervariable regions in K1 or other genes. Such protocols are more likely to detect errors that occurred during PCR. Our study found virtually no intra-sample or intra-host diversity even at K1 in the 9 individuals examined. While second enrichment could conceivably bias sequencing results, comparing doubly-enriched oral swabs libraries to their matching, singly-enriched tumor libraries resulted in identical sequences. Nevertheless, since recombination is evident in the evolutionary history of KSHV [36,38,46,50], co-infections by multiple KSHV strains have occurred, but this does not appear to be a common feature of KSHV infection.

At the point mutational level, sample-consensus KSHV genomes in tumors and oral swabs within the same individuals were nearly identical, differing by at most two point mutations. None of the differences occurred in introns and intergenic regions, and nearly all were nonsynonymous changes into dissimilar amino acids. Notably, the sole intra-host synonymous mutation observed occurred inside K12/Kaposin A of tumor U003-C (GK18 position 118,082), within the tumorigenic microRNA miR-K10a-3p [89]. The single nucleotide change, outside the seed sequence [90], is predicted to result in a slightly more stable stem loop precursor (ΔG -32.40➡-32.70, **S7 Fig**). The oral swab counterpart maintained the database consensus, and among published KSHV genomes, only tumor-derived ZM106 (GenBank KT271458) also had T at this position. The biologic implications of this change are not clear, but polymorphisms within this region have been reported to affect viral microRNA levels in KS lesions [91].

A striking finding in our study was the frequency of aberrant KSHV genomes in KS tumors, summarized in **Fig 9**. At least 4 of the 7 participants whose samples were examined had KSHV with major inversions, deletions or duplications in their tumors. In contrast, no aberrant

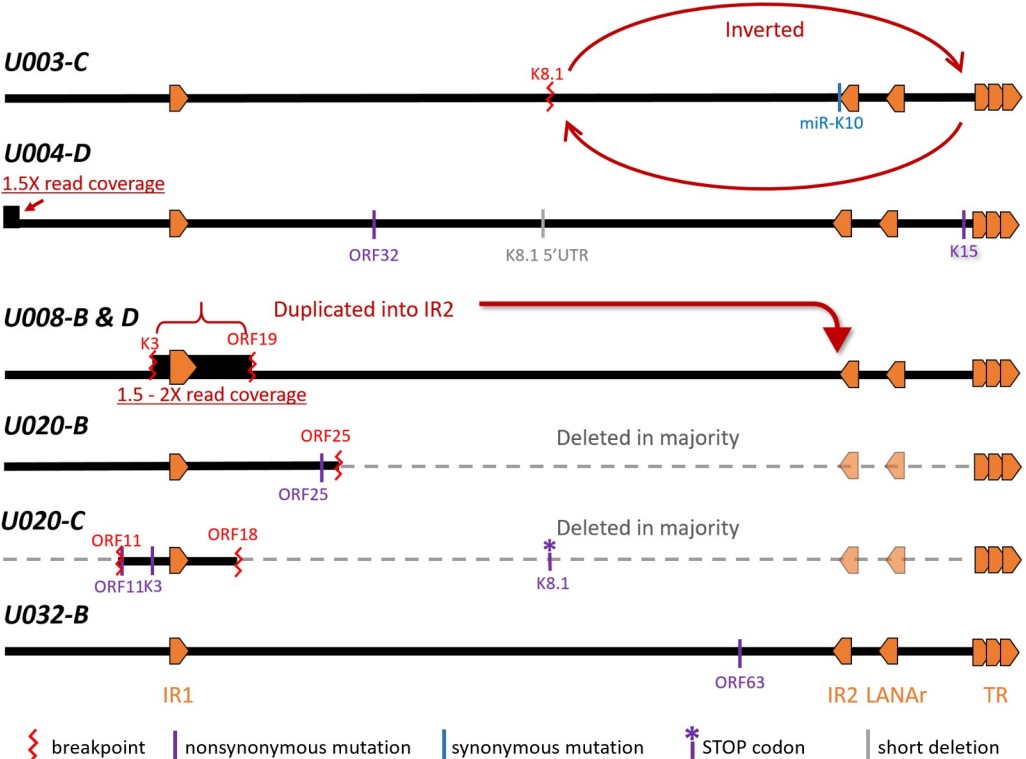

**Fig 9. Schematic representation of the 5 aberrant KSHV genomes discovered in KS tumors.** Specific details and evidence for each are referred to in the text and in Table 3. The nonsense mutation in K8.1 of U020-C were found in the remaining full-length genomes detectable in this tumor.

genome structures were found in oral swabs from the same individuals. Moreover, aberrant genomes comprised the majority of the KSHV genomes in the tumors in which they were detected, from 1.7-fold (U008-B) to 30-fold (U020-C) more aberrant genomes than full-length intact KSHV genomes (**Table 2** and **Fig 7A and 7B**). Further, no intact viral genomes were detected in 3 of 5 examined tumors from participant U003 (**Figs 5 and 8**). We did not find evidence of integration of KSHV into human chromosomal DNA; all breakpoints detected in KSHV sequence reads, when mapped, connected to other regions of the KSHV genome.

The fact that aberrant KSHV genomes were only observed in tumor samples is intriguing yet their significance is unclear, since it is well known that tumor cells suffer substantial genomic instability [92]. KSHV genome aberrations have been reported previously: The first whole genome sequence of KSHV published reported a 33-kb portion of the KSHV unique central region duplicated into the TR region [93]. A study of 16 tumor-derived KSHV whole genomes from Zambia reported 4 that had regions with as much as 3-fold more coverage than sample average, although the regions were not specified [33]. A PCR screen for some KSHV genes showed that some KS tumors and KSHV-infected B-cell lines can harbor deleted KSHV genomes [94], and one such B-cell line proliferated faster than the parental BCBL-1 line [94]. The infecting KSHV virus in this line had an 82-kb deletion from the 5' end of its genome, was lytic replication-incompetent, and could be packaged by a helper virus.

The genomic locations and character of tumor-associated mutations we observed suggest that mutations that propagated to high copy numbers may not have been random. However, again, this could represent regions of particular susceptibility to genome instability or they may contribute to tumorigenesis. All point and genomic mutations we observed impacted

protein coding sequences, and most rearrangement breakpoints truncated lytic gene products (**Tables 3 and** S3). Given that the KSHV genome densely encodes many immunomodulatory, angiogenic and anti-apoptotic factors [21–23], it is therefore possible that some mutations observed here could contribute to KS disease, or could involve viral genes that are unnecessary within established KS tumors, or contribute to immune evasion of transformed host cells.

The genomic region around IR1 featured prominently in genomic rearrangements in 4 tumors, potentially leading to their over-expression relative to other KSHV genes. For example, tumors U008-B and U008-D had a 14.8-kb portion of their genomes, from inside K3 to ORF19, duplicated into within IR2 (**Fig 6**). In a parallel RNAseq study, tumor U008-B had been found to abundantly express a chimeric transcript of the 14.8 kb section fused to IR2 sequences transcribed from a strong latency-associated promoter [81]. Distinct deletions were observed in tumors U020-B and U020-C from another participant, but the genomic regions retained, aside from TR sequences, again included the IR1 region (**Fig 7**).

IR1 is one of the origins of lytic replication, and transcripts around IR1 are among the most highly expressed in KS tumors [81]. These include two long non-coding RNAs that have indispensable roles during lytic reactivation of KSHV, T1.4 [68,95,96] and PAN [97–99]. PAN has been shown to interact with promoters of cellular genes involved in inflammation, cell cycle regulation and metabolism, and exogenous expression of PAN alone enhanced cell growth phenotypes [100]. Recently, virally-encoded circular RNAs encoded within PAN were discovered to be abundant in clinical samples and were inducible in KSHV-infected cell lines [101–103]. Other non-coding transcripts are potentially expressed from this region but their biological significance is unknown [99,104]. Finally, most ORFs encoded in the 14.8-kb retained region are lytic genes that have functions in subverting adaptive (K5/MIR2) [105,106] or innate immunity (K4/vCCL-2, K4.1/vCCL-3, K4.2 and K6/vCCL-1)[106–109], and apoptosis (K7 and ORF16/vBCL-2) [106,109], all of which might enhance the survivability of a host cell. There is evidence to suggest that KSHV lytic gene expression is crucial to KS pathogenesis [110], and that residual lytic gene expression plays a role in latent KSHV persistence *in vivo* [111], so it is possible that viral mutations in lytic genes could impact viral persistence or KS disease.

In addition to mutations with the IR1 region, the late lytic gene K8.1 was found to be mutated in KS tumors from 3 individuals (**Fig 9**). U003-C had an inversion breakpoint at the K8.1 intron, U004-D had a 28-bp deletion ending at 4 bases upstream of the first K8.1 exon, and the U020-C full-length minor genome had a nonsense mutation at the start of the second K8.1 exon. Furthermore, no intact K8.1 gene sequence was detectable by hemi-nested PCR in most tumors tested from participant 003 (**Figs 5C and 8**). Interestingly, truncations in K8.1 had been reported previously, and all were from KS tumor isolates. The original GK18 isolate has a 74-bp deletion at the 3' end (GenBank ID AF148805 K8.1 annotation); the Zambian isolate ZM124 (GenBank ID: KT271466) has a 25-nt deletion resulting in a frameshift and premature stop [33], and; Japanese isolate Miyako1 has a stop codon early in its first exon (GenBank ID LC200586 miscellaneous annotation). Gene K8.1 encodes an envelope glycoprotein that interacts with heparin sulfate for attachment [112–115]. It is not required for entry into endothelial [113] or 293 cells [116], although it had recently been shown to be necessary for infection of primary and cultured B-cells [117]. The K8.1 protein is among the most immunogenic KSHV proteins [118–120] and can be targeted by cellular immunity [121]. It is therefore conceivable that the preponderance of K8.1 mutations might be due to potent immune targeting of cells expressing K8.1 glycoproteins. Evading immune responses by omission of K8.1 expression may confer better survival of the host tumor cell. Altogether these findings suggest there could be selection pressure against K8.1 expression, which warrants further study.

That the same viral mutational signatures were found in separate KS lesions provides strong evidence that tumor-associated and likely defective KSHV genomes can spread by metastasis, or with a helper virus. Identical KSHV genome rearrangements in distinct tumors implies that the viruses were clonally related, since independently acquiring the same breakpoint coordinates is highly improbable. Participant U008 had 2 distinct tumors with the same 14.8-kb sequence duplicated into IR2 (**Figs 6 and** S5). Participant U003 had a KSHV genome inversion that would likely disable production of infectious virions, yet the signature K8.1-TR breakpoint junction was present in all his 5 examined tumors over 3 months (**Figs 5 and 8 and S2 Table**). Only 3 of the 5 had intact K8.1 sequence detectable by PCR (**Fig 8**). This suggests that the genomic inversion and K8.1 loss of function was not too detrimental to KS tumor persistence and implies a capacity for tumor-associated KSHV to continually seed tumors in other anatomic sites. To our knowledge, evidence of possible metastatic spread of KSHV genomes has not been previously reported, and these findings may have significant implications for understanding the progression of KS. However, consistent with the helper virus hypothesis, all deleted genomes retained the terminal repeat sequences, which contain the putative virion packaging (pac1) and cleavage sites.

In summary, highly accurate deep sequencing revealed that whole KSHV genomes in paired oral swab and KS tumors from individuals with advanced KS were virtually identical at the point mutational level. The use of dUMI provides a proof of concept for utilizing this technique to study other DNA viruses such as human cytomegalovirus, which can exhibit substantial intra-host genome heterogeneity [122]. Where there were differences, the viruses detected in saliva had the database consensus genotype while viruses detected in some tumors had novel mutations. We observed that KS tumors can harbor KSHV with genomic aberrations affecting coding regions, which raises the possibility that these mutations could alter the function of these genes and were of biologic significance. Moreover, our study demonstrated that specific KSHV genome aberrations can be found in distinct tumors from the same individuals, suggesting that infected tumor cells can seed other KS lesions. Future studies characterizing KSHV genomes in multiple tumors within individuals among a larger cohort of KS studies will help determine if tumor-associated KSHV mutations similar to those observed in our study result from tumor-associated genome instability, or contribute to KS tumorigenesis, or are associated with KS clinical presentation or clinical outcomes.

## Supporting information

**S1 Fig. dUMI-adaptors and primers for duplex sequenceing.** During library preparation, sheared DNA fragments were A-tailed and ligated with forked, double-stranded oligonucleotides containing Illumina TruSeq universal adaptor sequences, 12-random base pairs as dUMI and spacer sequences. The adapted DNA libraries were PCR amplified before enrichment with primers mws13 and mws20, which bind to Illumina Truseq adaptors. Primer mws21 containing sample index ID for multiplex sequencing was used for PCR following enrichment. DNA libraries post processing are shown at the bottom.
(TIF)

**S2 Fig. Workflow of genome assembly and variant analysis.** KSHV genomes were first assembled *de novo* from sequence reads of each sample, before being used as reference for mapping their respective dUMI-consensus reads (adapted from [61], see details in the Methods section). Discrepancies in bases between the sample-consensus genome and mapped dUMI-consensus reads were taken to be real intra-sample variants.
(TIF)

**S3 Fig. Read coverage in all reported samples.** Raw (light blue), sUMI (blue) and dUMI-consensus (dark blue) read coverage in log scale along the de novo assembled, sample-consensus KSHV genomes in tumors (A) and oral swabs (B) examined in this study. Major repeat regions were masked and seen here as no coverage regions.
(TIF)

**S4 Fig. Potential sequencing artifacts.** (A) KSHV intrasample variant frequency as a function of read coverage. Sample variant frequencies were shown in Table 1 when at least 100 viral genomes were sampled, since below that level, minor variant frequencies were judged to be unreliable. (B) Intra-sample minor variants detected in dUMI-consensus reads of all samples by type of base substitution.
(TIF)

**S5 Fig. U008-B and U008-D were from distinct lesions on the left leg.** U008-B biopsy was obtained from lesions in the upper thigh, while U008-D was biopsied from a large lesion on the knee.
(TIF)

**S6 Fig. Mutations of KSHV genomes in tumors from participants 004 and 003.** (A) Alignment of KSHV genomes from Participant 004, showing a 28-bp deletion in the K8.1 promoter in U004-D. U004-D and U004-C are from tumors while U004-o1 is from an oral swab. (B) The only intra-host synonymous mutation found in this study, within miR-K10 in participant 003. (C) Sequence chromatograms of miR-K10 in other tumors of participant U003, with a T in all tumors and a mixture of T and the database consensus C in a minority of viruses in U003-G.
(TIF)

**S7 Fig. Predicted secondary structure of the stem loop precursor of miR-K10a-3p.** The structure of pre-miR-K12-10a was predicted using *mfold* (http://unafold.rna.albany.edu/?q=mfold/RNA-Folding-Form), indicating the mature miRNA, seed sequence (grey) and the intra-host polymorphism (red) found in participant U003. The G➜A change in RNA sequence resulted in a slightly more stable stem loop (ΔG -32.40 ➜ -32.70).
(TIF)

**S1 Table. Adapter, primer and probe sequences used for library prep and ddPCR.**
(XLSX)

**S2 Table. Anatomic sites, biopsy times and tumor morphologies of the KS lesions.**
(XLSX)

**S3 Table. Tumor KSHV-unique mutations observed, in gene order.**
(XLSX)

**S4 Table. Number of dUMI-consensus reads from BCBL-1, tumor and oral swab samples mapping to EBV genome (strain B-95/AG876).**
(XLSX)

## Acknowledgments

We thank the study participants who contributed invaluable specimens that make this study possible. We also thank D. Depledge and J. Breuer for sharing their RNA bait sequences for capturing KSHV genomic sequences and M. Lagunoff for kindly providing us with an early passage BCBL-1 cell line and critical reading of this paper.

## Author Contributions

**Conceptualization:** Jason D. Goldman, Corey Casper, Warren T. Phipps, James I. Mullins.

**Data curation:** Jan Clement Santiago, Hong Zhao, Alec P. Pankow, Fred Okuku.

**Formal analysis:** Jan Clement Santiago, Alec P. Pankow.

**Funding acquisition:** Jason D. Goldman, Corey Casper, Warren T. Phipps, James I. Mullins.

**Investigation:** Jan Clement Santiago, Jason D. Goldman, Hong Zhao, Alec P. Pankow, Fred Okuku, Lennie H. Chen, C. Alexander Hill.

**Methodology:** Jan Clement Santiago, Jason D. Goldman, Hong Zhao, Alec P. Pankow, Michael W. Schmitt, Lennie H. Chen, James I. Mullins.

**Project administration:** Corey Casper, James I. Mullins.

**Resources:** Fred Okuku, Michael W. Schmitt, Corey Casper, Warren T. Phipps, James I. Mullins.

**Software:** Jan Clement Santiago, Alec P. Pankow, Michael W. Schmitt, C. Alexander Hill.

**Supervision:** James I. Mullins.

**Validation:** Jason D. Goldman, Hong Zhao, Michael W. Schmitt, Lennie H. Chen.

**Visualization:** Jan Clement Santiago.

**Writing – original draft:** Jan Clement Santiago.

**Writing – review & editing:** Jan Clement Santiago, Jason D. Goldman, Michael W. Schmitt, Corey Casper, Warren T. Phipps, James I. Mullins.

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
