## [Decision Letter · Decision Letter 0]

1 Jun 2020

Dear Dr. Mullins,

Thank you very much for submitting your manuscript "Tumor-specific changes in Kaposi sarcoma-associated herpesvirus genomes in Ugandan adults with Kaposi sarcoma" for consideration at PLOS Pathogens. As with all papers reviewed by the journal, your manuscript was reviewed by members of the editorial board and by three independent reviewers. The reviewers appreciated the novelty and technical proficiency of the study, particularly the use of the high depth dUMI technique and the within-individual comparison of KS tumor samples with those in the oral cavity. However, each reviewer raised significant concerns about the manuscript, including specific aspects of the study methodology and the way in which some of the results were interpreted and discussed. In light of the reviews (below this email), we would like to invite the resubmission of a significantly-revised version that takes into account the reviewers' comments. All of the issues deemed as major by the reviewers are, in my opinion, key issues that must be addressed in a revised manuscript, either with a more critical interpretation and discussion of the data and/or through performance of additional experiments. Several minor issues requiring attention were also highlighted. Reviewer suggestions/concerns that are recommended but not essential include modifying the title to more accurately reflect the study design, discussing the question of tumor metastasis in more detail, and revising the speculation about the role of K8.1 in tumorigenesis. 

We cannot make any decision about publication until we have seen the revised manuscript and your response to the reviewers' comments. Your revised manuscript is also likely to be sent to reviewers for further evaluation.

Sincerely,

Ashlee V. Moses

Associate Editor

PLOS Pathogens

Shou-Jiang Gao

Section Editor

PLOS Pathogens

Kasturi Haldar

Editor-in-Chief

PLOS Pathogens

orcid.org/0000-0001-5065-158X

Michael Malim

Editor-in-Chief

PLOS Pathogens

orcid.org/0000-0002-7699-2064

Reviewer's Responses to Questions

**Part I - Summary**

Reviewer #1: The study describes the duplex sequencing using unique molecular identifies (dUMI) of 12 matching oral and tumor samples were obtained from 9 Ugandan study participants, with HIV –associated KS. Libraries were enriched for KSHV using RNA baits of 120bp 12x tiling of the KSHV genome, previously described. Briefly, the RNA baits generate 5X coverage of the positive strand of the KSHV genome. The authors further enriched for K1, ORF75, K15, ORF26, and the TR sequences of several KSHV cell lines and western and African strain sequences. Several genomes had the presence of split reads, indicating aberrant genome structures, which were described in further detail and confirmed by PCR. Gene copy numbers of the K2, ORF16, ORF50/RTA, and ORF73/LANA genes were determined and reported to correlate with the observed shifts in read coverage. Variants calling was performed and reported to be non-synonymous, which were mostly inconsistent between the matching oral and tumor samples.

Reviewer #2: This is a technically adept report of the variability of KSHV whole genomes in comparison of human KS tumor samples to those in the oral cavity from the same individuals (in most cases). The novelty of the manuscript lies in the application of the high-depth dUMI method and the resulting high-accuracy base calls and description of intra-host variation that has previously only focused on tumor samples from the same subject and never been reported at this depth. The findings of genomic aberrrations in tumor-derived but not in the oral samples are quite interesting but the results are perhaps over-interpreted as being tumor drivers (lines 51-52 and elsewhere) as opposed to a consequence of the genomic instability inherent in tumor microenvironments that may adversely impact the stability of KSHV episomes in the course of their replication and partitioning to daughter cells. Moreover, the finding that a signature aberration was found in more than one tumor is strong evidence for metastasis of a KS tumor, in the likely absenece of the capacity for the etiological agent to spread via an extracellualr virological mechanism. This reviewer felt this was an important finding that was curiously not highlighted to any significant degree in the manuscript. Because genomic aberration would not necessarily impact the entirety of the collection of KSHV episomes that typically persist in each infected cell, the results are consist with some of the viral genomes being altered and others being intact and the authors did mention that expression from intact genomes might complement the proagation of the defective genomes.

In light of previous findings, the justification for doing the work and for doing the work at the incredible level of sophistication was somewhat lacking given that the outcomes largely showed what might have been anticipated from the literature, with the exception of the genomic duplications, inversions and deletions. But as stated above, the significance of these in pathogenesis remains debatable.

There was some confusion in the presentation of the number of samples being analyzed versus those actually being being compared tissue to tissue. See Line 31 versus 107-108

In the PCR experiments designed to interrogate the genomic aberrations and identify the breakpoints by Sanger sequencing there are instances where one side of the junction is clearly identified and the other is only evident from the deep sequencing method. While this reviewers interpretation of the findings is largely in agreement with the authors', there is not real explanation offered as to why one junction is readily detect and another is not since they should be equimolar in the sample amplification reactions.

Reviewer #3: The manuscript described the whole KSHV genome analysis from KS tumor lesions of 9 Ugandan patients and 7 with paired saliva KSHV sequences for a total of 23 specimens analyzed by next generation sequencing. The results showed that there are very little intra-host sequence variabilities and there is no evidence of KSHV quasispecies or super infection within the same patient. This is in contrast to some earlier studies with analysis of specific KSHV genes, showing KSHV sequence variability and tissue compartmentalization, and will be important to document such findings. The authors also showed that there are however point mutations, sequence deletions, and gene rearrangements in the KS tissues in some of the patients, and in some cases in more than one lesion but not all the lesions analyzed within the same patients. This study in general was quite well done; the sequencing was carefully performed using the duplex sequencing with dUM1 tag, with very low sequence errors, and only unique viral template was sequenced to avoid duplications. This paper is a follow up of the RNAseq of the KSHV mRNA obtained from the same patients by the same group published recently. The study however, focused more on the technical details of sequencing and sequence analysis but less on the biology and significance of the findings. There are no major new findings but this sequencing method validates some of the previous findings like single KSHV subtype per person by Rose et al. Further discussion with emphasis on whether these mutations in KSHV caused tumorigenesis, or whether the rapid replication of cancer cells caused mutations in KSHV genomes, will be helpful.

**Part II – Major Issues: Key Experiments Required for Acceptance**

Reviewer #1: Major comments

1. The study could benefit enormously from including an expert in KSHV and herpesviruses. For instance, the premise that different subtypes of KSHV are associated with different disease manifestations is far from certain (line 79). The EBV data cited in the introduction, likewise, are still widely debated. The field has been down to many false paths due to overblown claims of that nature. The authors say as much on line 88.

Likewise, it is not at all clear if genome recombination is a consistent phenotype of KSHV or an artifact of lab culture. Recombination is rare and not normally associated with disease scores in any of the herpesviruses.

1. The authors overstate the significance of unique molecular identified for DNA sequencing and variation discovery of DNA material. The myriad of FDA-approved tests in cancer as well as other studies show that in the case of DNA, sufficient read depth can overcome many of the concerns raised in the introduction. Errors due to NGS methods play a much larger role in sequencing RNA viruses, as reverse transcriptase has a much higher error rate than modern DNA polymerase used for NGS.

2. It would have been better if in a manuscript about KSHV more relevant references about KSHV were provided and less for EBV and HIV. The dUMI technique and related approaches are sufficiently developed and now commercially available. There is no need to go into extensive detail about it.

3. Sequences need to be submitted to GenBank before acceptance.

4. The use of dUMI is novel and truly instills confidence in the analysis. It suggests that in the control ~ 5 reads give rise to one dUMI and above further collapse 50 raw reads per dUMI (line 251). How do the authors explain the difference in tumor vs oral dUMI coverage? The tumor samples all seem to work very well, but the majority of oral samples failed to yield satisfactory dUMI coverage, even though raw read coverage was similar to tumor coverage?

5. There is an alternative explanation to breakpoints not supported by junction reads, at least theoretical. Any KS tumor includes full-length latent genomes, perhaps defective genomes, perhaps integrated genomes, but also encapsulated virion DNA and, defected, replication intermediates. The proportion of these varies depending on the number of lytic cells in the lesions.

6. It is in the discussion where this paper veers off course. The authors did not generate ">100 error-free genomes" line 444, but only 12 for 12 tumors. The majority of oral samples have less than sufficient mean coverage.

7. The conclusion that the observed intra-tumor rearrangements were not random (line 465) is premature. First, the limited number of samples does not allow for a statistically significant answer to this hypothesis. Second, during latency any mutation outside the genes required for latent persistence is neutral. While mutations that interfere with latent replication are selected against at every cell division. It is premature to speculate on the oncogenic benefit of any of the observed rearrangements (line 470) or specific selection pressures against K8.1. (line 509).

Reviewer #2: The authors should consider moderating the implications of the genomic variation or genomic aberration of viral pathogenesis since it is not clear that either variation or aberration is driving disease, as opposed to a consequence of it.

In lines 92 - 94 the authors mention that intertypic recombinants exist suggesting that co-infection of divergent strains must occur, but in fact it would seem to require co-infection of single cells with differential genetic subtypes. it is unclear whether the sequencing data presented provides evidence of such intracellualr KSHV diversity from a single cell to support the potential for forming intertypic recombinants. This comes up again in the Discussion Lines 458-459- is it co-infection of a single cell or leading to recombination, or accumulation of KSHV episomes leading to some sort of replication error by the cellular replication machinery trying perhaps to resolve the repeat structures in the KSHV episomes? It would seem that the probability of such, otherwise unlikely, events would increase with the episomal burden in cells.

Line 460-462- This reviewer agrees that the genomic aberrations are profoundly interesting and COULD be important causally, but it was unclear (perhaps my fault) what the frequency of such aberrations were of the total KSHV genomes from within a given tumor harboring those mutations. Is this frequency variable and does it associate with any other parameters of KSHV or HIV-1 co-infection- for example tumor morphotype.

Figures 5 and 6; Fig 5-Are the anticipated product sizes for amplicons (4 and 5) deemed to be diagnostic of the rearrangement and duplication correctly calculated because none seem close to those on the Agarose gels accept 1, 2, and 3.

Fig 6- The absence of seemingly critical confirmatory PCR products confounds the appreciation and absolute confirmation of the HTS data. For example Reaction 3 and 4, no product? So it is all based on the interpretation of reaction 5. The figure legend suggest the bands were excised and Sanger sequenced, but there don't appear to be diagnostic bands. So it would seem that one junction is a 'known and the other is implied from but incompletely define

Reviewer #3: 1. Figure 4 C and D, it is not clear why 4D used BCBL and not U0007B as control, as in 4C?

2. For U003 there are a number of protein altering changes in the saliva sequence as compared to the tumor sequences, where are these changes, how many of these changes are common and where do they differ, and how do they affect the proteins’ functions?

3. Why did lane 4 of Fig 6G lane 4 did not show the 1kb PCR product if K2 is linked to TR as expected, while lane 5 did show a strong band that flanks the 3’ insertion site?

4. Fig 5S phylogenetic analysis is important since it compares the 9 KS sequences analyzed with the currently existing whole KSHV genome sequences, thus it should be shown as a regular figure, Moreover, the authors should comment on the UG156, why it is more closely related to the KSHV sequences from the western countries.

5. The KSHV sequences obtained are from homogenized tumor lysates, so how can the authors be certain that the final assembled KSHV sequence represent the actual viral genome in the tumor cells, whether multiple genotypes are within a single cell or in different tumor cell populations, or even are from extracellular viral particles.

6. The authors has focused the discussion on the significance of the role of alternation of the KSHV sequences in KS tumorigenesis and potential immune escape mechanisms by the virus. However, more emphasis should be placed on tumor DNA sequences instability as a result of rapid tumor cell growth and tumorigenesis, and may not have anything to do with viral pathogenesis and tumorigenesis. Moreover, it has been shown that KS tumor cell cultures without detectable KSHV sequences can be established. Thus, changes of KSHV sequences could occur after tumor establishments.

7. The authors have suggested that since K8.1 can generate an effective immune response and thus alternations in K8.1 gene could provide an advantage for viral escape, pathogenesis and tumor growth. However, it has been shown that KS patients still mount a robust humoral immune response against KS, including against K8.1. Thus, it is unlikely that changes in K8.1 paly an important with KS tumorigenesis.

8. The result showing that in some cases more than one tumor lesion in the same patient harbors the same KSHV alternations is interesting. This suggests that KS lesions could be clonal origin due to metastases, but could also arise independently. Such points should be emphasized in the discussion.

**Part III – Minor Issues: Editorial and Data Presentation Modifications**

Reviewer #1: 1. Minor comments

2. On line 253, the authors reference Figure 2B, which does not exist.

3. The author’s reference a previously described duplication in BAC-36 in line 261, and have incorrectly reported the duplication size as 19kb, instead of 9kb.

4. Table 1, page 13. The table is not formatted properly and indicates a missing field on the left-hand side. The mean read coverage seems to represent the number of reads and maybe misleading when looked at with supplementary Figure 3, which reports read coverage. The authors don’t describe if data was normalized. Please use the median instead of mean coverage and provide 95%CI or sd in table 1.

5. The Authors describe the inversion of the K8.1 exon in Tumor U003C, Line 330. The legend in Figure 4, describes the presence of the faint band seen in the BCBL-1 control TR-K8.1, Fig. 4D, as the presence of the forward primer amplicon. Perhaps different primers can be designed to confirm the band/or lack of a band in the control sample. It does not clearly state that the bands found in the control sample were confirmed by Sanger sequencing to be the forward primer amplicons. The text describing the data in Figure 4C in line 335 is different from the figure legend. The Figure and its legend indicate that the samples being compared are from different individuals as opposed to the same patient, as described in the text. There is a Typo in line 330, A ‘U’ is missing in the tumor sample number.

6. Line 368, the authors describe the breakpoints inside ORF11 and ORF18 and fusion into the TR. PCR primers are designed and confirm the 3’ fusion of ORF18 and the TR but not the fusion of ORF11 and the TR at the 5’ end, Fig. 6F, row 4. This is also shown in Fig. 6G, lane 4. The authors did not explain the presence of faint bands in Fig 6G, lane 7, 8, and 9, which may indicate the presence of these PCR products. Figure 6F, lane 5 referenced in line 377 is the same PCR product as the 3’ junction product reported in line 374, Figure 6G, lane 5. The wild type product of lane 2 in Figure 6F and G is much larger than the expected band size. In conclusion, no PCR confirmation was provided to verify this finding.

7. It could be argued that mutations are called when a low number of reads obtained, despite selecting mutations with a frequency of 100. As mentioned in their references, these mutations could be a consequence of DNA damage and other cellular events.

8. The resolution of the Figures could be improved on.

9. (line 301) If variants were supported by only one read, where these excluded from the consensus sequences submitted to GenBank?

10. line 417: Did this SNV disrupt the target sit of miR-K12-10? Please provide a mfold structure comparison.

11. line 439: Would this technique detect HIV, since it does not use reverse transcriptase or was HIV assayed by viral load assay?

12. Please remove line 521 "or with a helper virus" as there is no evidence for reinfection.

Reviewer #2: Some discussion of why a secondary library enrichment of low copy# samples does not bias the sequencing outcomes.

LINE 247- POLG not defined

LINE 275- How does going from a median of 0.17% to 1.3% equal a 6000-fold increase?

LINE 466-469- Brings up a good concept but sufferer from the lack of comparison to data from immunological studies to suggest the immune response to the gene products indicated are associated with protection against KSHV disease. The authors should discuss since Whitby and others have conducted whole proteome B and T cell repertoire screening without defining patterns of immunodominance or correlates of protection.

LINE 480- RNASeq or transcriptomic/array data, in this reviewer's experience, seems to suggest that there is no single pattern of KSHV expression associated with tumors. Almost exclusively lytic exclusviely latent, and mixed patterns have been detected in association with similarly staged tumorigenesis, so I not clear as to the accuracy of the statement.

LINE 489-491-This seems like a list of genes and functions without a clear linkage to the discussion above it. Is the suggestion from the RNASeq data from the same group demonstrating up or down regulation of these genes in concert with the IR1 aberration? The conclusion is not evident and could be clarified.

LINE 932-933- first statement is inaccurate since not all subject provided both samples.

Reviewer #3: Table 2 title needs to be changed. It titled as gene copy in tumor DNA but the table showed oral KSHV sequences which are not from tumors.

PLOS authors have the option to publish the peer review history of their article (what does this mean?). If published, this will include your full peer review and any attached files.

Reviewer #1: No

Reviewer #2: No

Reviewer #3: No
---

## [Decision Letter · Decision Letter 1]

26 Oct 2020

Dear Dr Mullins,

Thank you very much for submitting your manuscript "Intra-host changes in Kaposi sarcoma-associated herpesvirus genomes in Ugandan adults with Kaposi sarcoma" for consideration at PLOS Pathogens. Your revised manuscript was reviewed both by members of the editorial board and two of the reviewers who considered the initial submission. The reviewers felt that the revised manuscript had successfully addressed most of the prior concerns, but felt that a more substantive discussion about the findings of your study and their implications was warranted. Most notably, please discuss in more depth the possibility that the deletions, duplications and changes detected in the tumor cells could be due to the genome instability of the cells, rather than being the cause of tumorigenesis. In addition, please address the query about the figure designation on line 265. Based on the reviews, we are likely to accept this manuscript for publication, providing that you modify the manuscript according to the review recommendations.

Sincerely,

Ashlee V. Moses

Associate Editor

PLOS Pathogens

Shou-Jiang Gao

Section Editor

PLOS Pathogens

Kasturi Haldar

Editor-in-Chief

PLOS Pathogens

orcid.org/0000-0001-5065-158X

Michael Malim

Editor-in-Chief

PLOS Pathogens

orcid.org/0000-0002-7699-2064

Reviewer Comments (if any, and for reference):

Reviewer's Responses to Questions

**Part I - Summary**

Reviewer #1: The authors have addressed the reviewers' concerns.

Reviewer #3: This is a highly technical but interesting study on a unique set of KS patients’ specimens, which include matching saliva and tumor tissues from the same patients. Some with multiple tissue biopsies. There were 7 matched saliva and tumor tissues analyzed, and 4 individuals have more than 1 tumor lesions characterized. The authors showed that there were essentially no intra-patient variations among the sequences analyzed within the same individual, in contrast to some previous studies which have analyzed part of the KSHV genomes and demonstrated intra-patients’ variations. The authors here found that there were gene duplications, inversions and mutations in the tumors from 4 out of the 7 cases. The strength is that the whole KSHV genome sequencing used the molecular identifiers (dUMI) to barcode individual DNA molecules before library amplification so minimize PCR generated mutations. The study was also meticulously conducted and analyzed. The indications of reversions and duplications based on the number of reads were carefully confirmed and analyzed. The revised manuscript has also addressed most of the previous reviewers concerns. The concerns were mostly on the over interpretation of the findings based on the limited cases analyzed. This revision has tempered down most of the discussion on the mechanistic implications of the findings and the authors have made changes to the text. However, there are a couple of points that the authors could further consider.

1. This reviewer agrees with the previous reviewers that the findings on the deletions, duplications and changes are most likely due to the result of genome instability of the tumor cells rather than the cause of tumorigenesis. Similar changes in multiple lesions can be the results of the metastases of tumor cells harboring the mutations in the KSHV genome. This was demonstrated by case U003 that only 3 out of 5 lesions have the mutations and the other two did not, and were similar to the saliva KSHV sequences. This suggests that KS can develop in the absence of changes in the KSHV genome, and changes have likely occurred after tumorigenesis, and thus KSHV genome changes may not have substantial influence on pathogenesis. Further discussing this possibility will be helpful.

2. As one of the reviewer has pointed out “the justification for doing the work and for doing the work at the incredible level of sophistication was somewhat lacking ….” Further strengthening the rationale will be useful.

3. Line 265, are the authors referring to Fig. 1B rather than Fig. 3B?

**Part II – Major Issues: Key Experiments Required for Acceptance**

Reviewer #1: NA

Reviewer #3: (No Response)

**Part III – Minor Issues: Editorial and Data Presentation Modifications**

Reviewer #1: The conflict of interested statement should be expanded upon to explain that the authors company expressly markets the technology used in this manuscript.

Reviewer #3: (No Response)

PLOS authors have the option to publish the peer review history of their article (what does this mean?). If published, this will include your full peer review and any attached files.

Reviewer #1: No

Reviewer #3: No
---

## [Editor Report · Decision Letter 2]

24 Nov 2020

Dear Dr. Mullins,

We are pleased to inform you that your manuscript 'Intra-host changes in Kaposi sarcoma-associated herpesvirus genomes in Ugandan adults with Kaposi sarcoma' has been provisionally accepted for publication in PLOS Pathogens.

Best regards,

Ashlee V. Moses

Associate Editor

PLOS Pathogens

Shou-Jiang Gao

Section Editor

PLOS Pathogens

Kasturi Haldar

Editor-in-Chief

PLOS Pathogens

orcid.org/0000-0001-5065-158X

Michael Malim

Editor-in-Chief

PLOS Pathogens

orcid.org/0000-0002-7699-2064
---

## [Editor Report · Acceptance letter]

13 Jan 2021

Dear Mullins,

We are delighted to inform you that your manuscript, "Intra-host changes in Kaposi sarcoma-associated herpesvirus genomes in Ugandan adults with Kaposi sarcoma," has been formally accepted for publication in PLOS Pathogens.

Best regards,

Kasturi Haldar

Editor-in-Chief

PLOS Pathogens

orcid.org/0000-0001-5065-158X

Michael Malim

Editor-in-Chief

PLOS Pathogens

orcid.org/0000-0002-7699-2064